# Life-history stage determines the diet of ectoparasitic mites on their honey bee hosts

Bin Han [1,4], Jiangli Wu [1,4], Qiaohong Wei [1,4], Fengying Liu[1], Lihong Cui[2], Olav Rueppell [3] ✉ & Shufa Xu [1] ✉

Ectoparasitic mites of the genera *Varroa* and *Tropilaelaps* have evolved to exclusively exploit honey bees as food sources during alternating dispersal and reproductive life history stages. Here we show that the primary food source utilized by *Varroa destructor* depends on the host life history stage. While feeding on adult bees, dispersing *V. destructor* feed on the abdominal membranes to access to the fat body as reported previously. However, when *V. destructor* feed on honey bee pupae during their reproductive stage, they primarily consume hemolymph, indicated by wound analysis, preferential transfer of biostains, and a proteomic comparison between parasite and host tissues. Biostaining and proteomic results were paralleled by corresponding findings in *Tropilaelaps mercedesae*, a mite that only feeds on brood and has a strongly reduced dispersal stage. Metabolomic profiling of *V. destructor* corroborates differences between the diet of the dispersing adults and reproductive foundresses. The proteome and metabolome differences between reproductive and dispersing *V. destructor* suggest that the hemolymph diet coincides with amino acid metabolism and protein synthesis in the foundresses while the metabolism of non-reproductive adults is tuned to lipid metabolism. Thus, we demonstrate within-host dietary specialization of ectoparasitic mites that coincides with life history of hosts and parasites.

The global decline of insect pollinators has become a serious threat to biodiversity and food security[1–4]. Ongoing colony losses of the Western honey bee (*Apis mellifera*), a model species and key pollinator in many natural and managed ecosystems, are of particular concern[5–8]. Threats to honey bee health include pathogens, agrochemicals, pollutants, dwindling food sources, and climate change[9–12]. However, the ectoparasitic mite *Varroa destructor* (Mesostigmata: Varroidae, Fig. 1a and b) is often identified as the most important contributor to honey bee health problems[13–15]. *V. destructor* weakens host fitness directly by feeding and indirectly by transmitting and potentiating viruses[16–18]. Another honey bee ectoparasitic mite, *Tropilaelaps mercedesae* (Mesostigmata: Laelapidae, Fig. 1c and d), coexists with *V. destructor* in *A. mellifera* colonies in most Asian countries. While *V. destructor* has

spread worldwide after its host-shift from the Eastern honey bee (*Apis cerana*), *T. mercedesae* is still limited to Asia after its host shift from the giant Asian honey bee species (*Apis breviligula, Apis dorsata*, and *Apis laboriosa*) about 50 years ago. Thus, our understanding of *Tropilaelaps* spp. is limited, although at least two species have the potential to become a more virulent honey bee pest than *V. destructor*[19,20]. Extensive efforts to develop chemical control of either species with miticides, organic acids, and essential oils, biological control via entomopathogenic fungi, engineered symbionts, and RNA interference, management control, selective breeding, or combinations of these approaches are ongoing[21–23] and rely on detailed knowledge of the mites' biology, which we are still lacking.

[1]State Key Laboratory of Resource Insects, Institute of Apicultural Research, Chinese Academy of Agricultural Sciences, Beijing 100193, China. [2]Cell Biology Facility, Center of Biomedical Analysis, Tsinghua University, Beijing 100084, China. [3]Department of Biological Sciences, University of Alberta, Edmonton, Alberta T6G2L3, Canada. [4]These authors contributed equally: Bin Han, Jiangli Wu, Qiaohong Wei. ✉e-mail: olav@ualberta.ca; xushufa@caas.cn

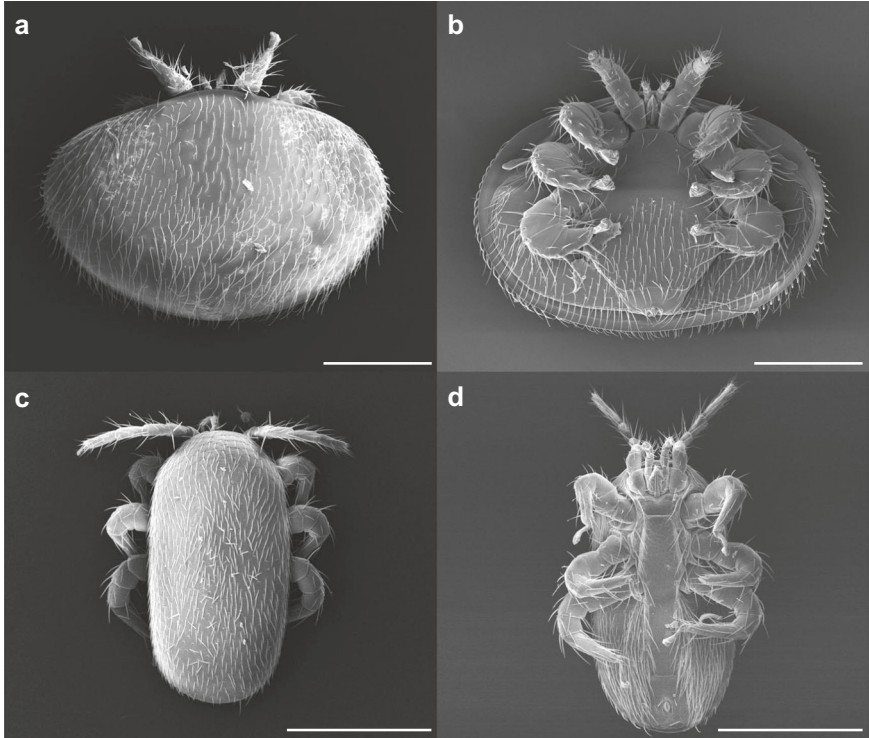

**Fig. 1 | Scanning electron microscope images of adult female bee mites.** Dorsal (**a**) and ventral (**b**) views of *Varroa destructor* compared to dorsal (**c**) and ventral (**d**) views of *Tropilaelaps mercedesae* indicate many similarities between the two ectoparasites despite the overall size and body shape differences. Both species have expanded their host use to *Apis mellifera* and threaten honey bee health. All scale bars represent 0.5 mm.

The life cycles of both mites are generally similar and intimately linked to their honey bee hosts, although only *Varroa* spp. alternate between clearly distinct dispersal and reproductive stages. The entire reproductive stage is performed inside a brood cell in the honey bee colony and starts with a mature female mite (foundress) invading the cell just before it is sealed by nurse bees. While *V. destructor* foundresses establish one feeding site that she and her offspring will share throughout development, *T. mercedesae* use multiple feeding sites on one host[24]. In both species, the foundress produces an initial haploid male egg followed by a series of female eggs and all offspring grow rapidly during a protonymph and deutonymph stage on the same host. During this stage, the foundress and her offspring exhibit well-organized behavior to maximize reproductive success despite limited time and space in the brood cell[25–27]. After maturing from the protonymph and deutonymph stages, the adult male and female offspring mate with each other before females emerge from the brood cell with their developed honey bee host to disperse and find another host to initiate the next reproductive cycle[13,27].

Compared to *V. destructor*, *Tropilaelaps* spp. mites have evolved a faster life history[19,28]. The *Tropilaelaps* foundress starts laying eggs about 10 hours after capping, as opposed to 60 hours in *V. destructor*, potentially because *Tropilaelaps* does not need to feed on host brood to stimulate oogenesis[29]. *Tropilaelaps* also lay eggs at a faster rate, at about 24-hour intervals[19], compared to 30-hour intervals in *V. destructor*[30]. The dispersal stage of *Tropilaelaps* spp. lasts only about 1.3 days and does not involve feeding on adult hosts. Instead, they might feed on unsealed larvae[29] because their mouthparts and body shape are not suitable for feeding on adult bees[19,31]. In contrast, *V. destructor* mites remain on adult hosts for usually 4–6 days when brood is available and for several months when brood is absent during the winter, feeding on adults through the intersegmental membranes of the abdomen during these periods[32–34]. The resulting higher reproductive rate and shorter vulnerable dispersal stage of *Tropilaelaps* spp.

mites facilitate a rapid population buildup, potentially presenting an even greater risk for honey bee health than *V. destructor*[20].

The regulation of reproduction and development of *V. destructor* is partially understood: Proteomic analyses suggest that enzymes involved in glycolysis and the citric acid cycle support the unique energetic requirements of the foundress to initiate reproduction, whereas proteins related to chitin metabolism contribute to the formation of the exoskeleton during development[35]. A full life cycle transcriptomic profiling of foundresses showed that energetic and chitin metabolic processes are more active in the dispersal stage, while transcriptional regulation predominates during egg laying[36]. Our recent lncRNA-mRNA co-expression analysis also identified a stage-specific expression pattern during development and proposed the deutonymph as a key stage during which energetic metabolism, metamorphosis, and detoxification are prominent[37]. Comparatively little is known about the distinction of life history stages of *Tropilaelaps* spp., except that its genome bears specific features that suggest a simplified behavioral repertoire under a dark and stable environment[38], reflecting a life cycle that is predominantly performed inside the honey bee brood cell.

In a paradigm-shifting study, Ramsey and colleagues changed the widely-held view of *V. destructor* feeding on hemolymph[13] by demonstrating with different fluorescent stains that it primarily feeds on the fat body when parasitizing adult bees[33]. Their subsequent physiological experiments showed that artificially increasing the amount of fat body in *V. destructor*'s diet can increase its survivorship and fecundity[33]. However, decisive experiments on the feeding behavior of the reproducing foundress mite and her offspring are lacking and we hypothesized that they differ in their primary food source while feeding on honey bee pupae. Importantly, the body of honey bee pupae fundamentally differs from that of adults. While the adult fat body is concentrated in a sheet of tissue underneath the abdominal cuticle, the fat body of the metamorphosing pupae is diffuse in the

hemolymph-filled body cavity during the remodeling and formation of adult tissues[39,40]. These strong differences in resource availability associated with specific life history stages of the host are predicted to result in different host utilizing strategies[41], which we aimed to empirically test in this study.

Here, we show in a series of experiments that the ectoparasitic mites *V. destructor* and *T. mercedesae* feed primarily on hemolymph during their reproductive life-history stage while parasitizing honey bee brood. We relate feeding site location with host tissue location during host development, demonstrate biostained hemolymph accumulation in the mites, and determine that host proteins in the mites are mostly derived from the hemolymph. We complement these results with additional proteomic and metabolomic analyses to characterize the physiological consequences of hemolymph feeding in *V. destructor*.

## Results and Discussion

The primary aim of this study was to distinguish between hemolymph- and fat body-feeding in honey bee mites while they parasitize the pupae instead of adults. Thus, we first performed a series of experiments on *V. destructor* to complement the results by Ramsey et al.[33]. We identified the sternite of the second abdominal segment as the major feeding site of foundress *V. destructor*. In conjunction with the absence of fat body in the honey bee host at this time in this location, these results led us to hypothesize that *V. destructor* use hemolymph instead of fat body during their reproductive foundress stage. Based on this hypothesis, we tested the prediction that the feeding of *V. destructor* on biostained honey bee pupae results in higher transfer of hydrophilic Uranine (staining hemolymph) than lipophilic Nile red (staining fat body). We further tested the prediction that honey bee-derived proteins in foundress *V. destructor* were distinct from such proteins found in dispersing females and contained disproportionally more proteins found in the honey bee hemolymph. To complement this evidence, a metabolome analysis was used to test the predictions that dispersing and foundress adult *V. destructor* rely on distinct physiology and that foundresses are overall more similar to juveniles that share their food source than to dispersing adults. We added comparative biostaining and proteomic analyses of *T. mercedesae*, ectoparasitic bee mites that have a strongly reduced dispersal stage and exclusively feed on honey bee brood, to support our finding that

brood-feeding mites predominantly access the hemolymph of their honey bee hosts. Taken together, we propose a life stage-specific food theory for *V. destructor*: Adult mites feed mainly on the fat body of adult bees during the dispersal stage, while adults and offspring rely primarily on the hemolymph of honey bee pupae during the reproductive stage in the capped brood cells.

## Feeding sites of *Varroa destructor* mites on honey bee pupae

We first investigated the feeding site of *V. destructor* mites on honey bee pupae because the source of nutrition for parasites can be largely determined by the site of parasitism on the host. Using vital staining, we located the position of the feeding sites of *V. destructor* on 1285 pupae, most of which ($n = 1266$, 98.5%) had only one integumental wound, while the remaining 1.5% of pupae ($n = 19$) had two perforations. We thus confirm quantitatively that the foundress mite prepares a unique feeding site on the pupal host for herself and her offspring[25]. This location contrasts with the typical feeding site of dispersing *Varroa* spp[33]. and the multiple feeding sites of *Tropilaelaps* spp. on honey bee pupae (see below). The feeding site is kept open by anticoagulating mite saliva proteins[42] and can thus be used throughout the mite life cycle[25]. Although our results cannot exclude additional feeding sites that mites initiate on older pupae, the energetic cost for the mite[25] and potential for accidental host mortality[26,43] would make such behavior maladaptive and thus unlikely.

With 90.0% of all observed perforations, the abdomen is the preferred feeding location of *V. destructor*. The majority of feeding sites ($n = 1101$, 85.7%) were established on the sternite of the 2nd abdominal segment (Fig. 2a and b). The strong preference of *V. destructor* for the second abdominal sternite corroborates earlier findings[44,45] and may be due to a relatively thin cuticle covering a large space containing hemolymph[44]. Thus, a large amount of hemolymph might be accessible with relatively little effort to penetrate the cuticle. The strong *V. destructor* feeding preference for the second abdominal segment suggests that this feeding site is beneficial for the mites but no data links the feeding site to offspring number so far. The small number of mesothoracic (10.0%, Fig. 2c and d) and other abdominal feeding sites between the 3rd and 7th tergite (4.3%, Fig. 2e and f) presumably represent instances when foundresses cannot access their preferred site[25]. Together, these results suggest that the majority of our subsequent analyses of reproductive foundresses involve

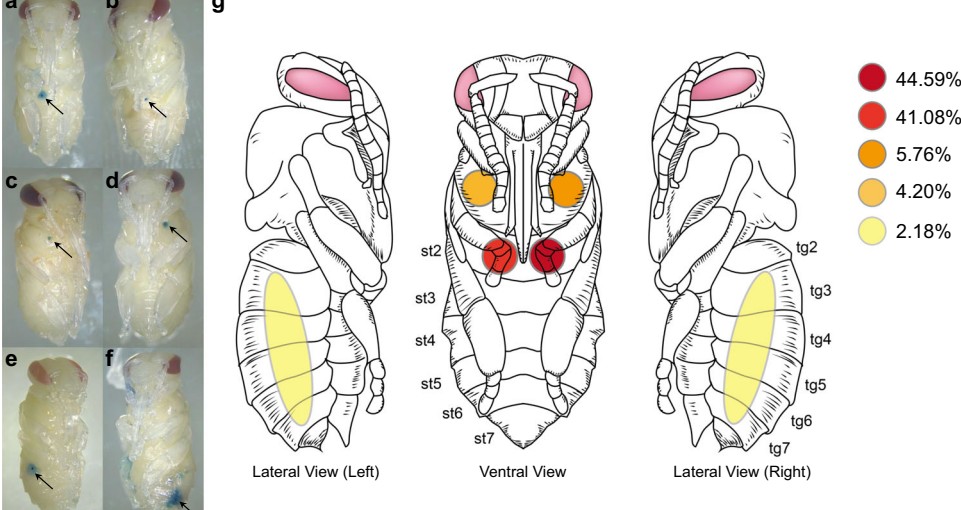

**Fig. 2 | *Varroa destructor* shows a strong preference for feeding on the 2nd abdominal segment of honey bee pupae. a–f** Representative photos of different *V. destructor* feeding sites, detected by vital staining of integumental wounds in different parts of the pupa. Each arrow indicates the location of a perforation. 1285 pupae were inspected in eight trials. **g** Diagram showing the frequency of integumental wounds found in each location on 1285 pupae (st, sternite; tg, tergite). Source data are provided in the Source Data file.

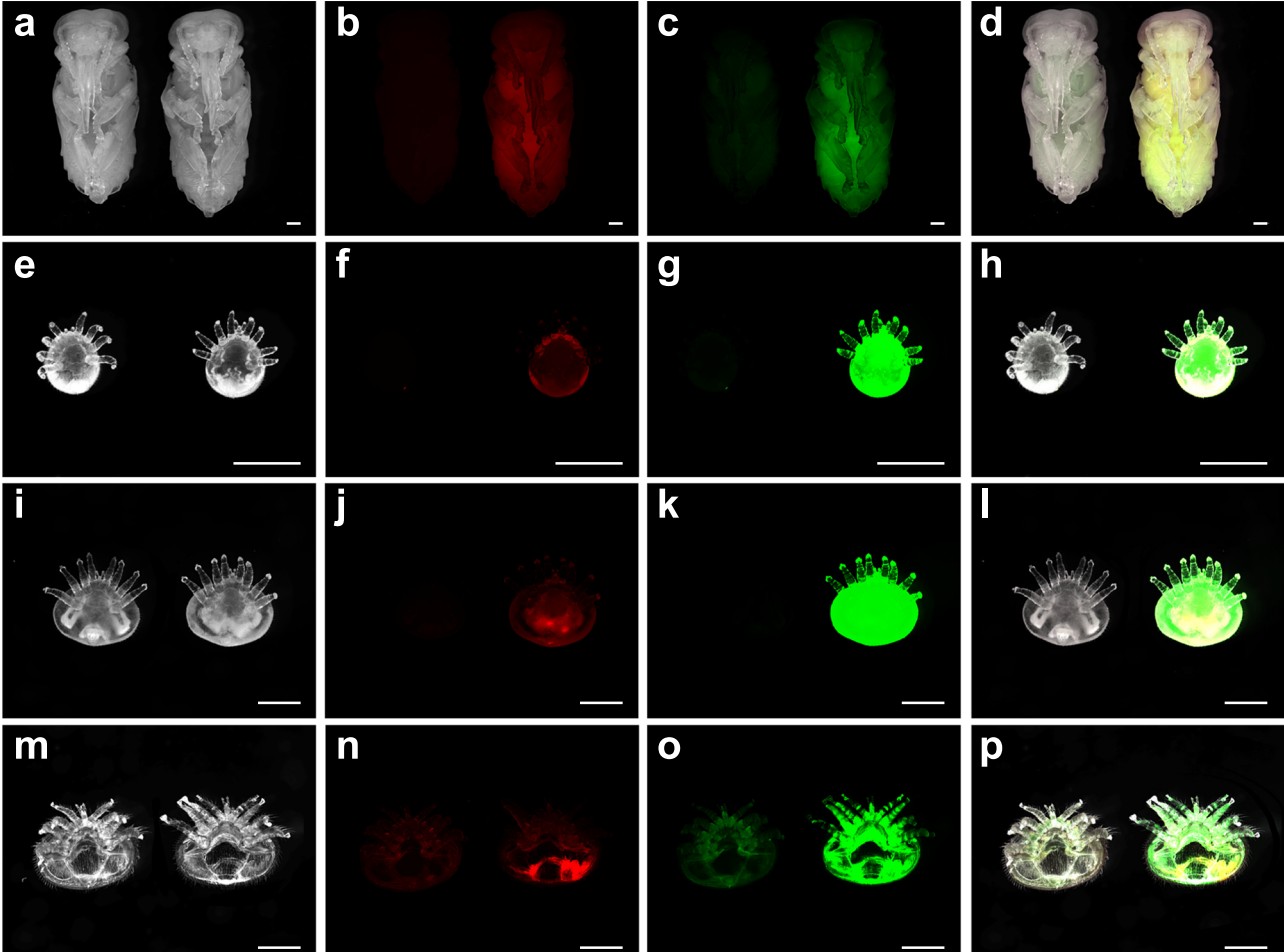

**Fig. 3 | *Varroa destructor* ingests large amounts of honey bee pupae hemolymph during their reproductive stage.** Photos show honey bee pupae that fed as larvae on biostains and *V. destructor* mites fed on these biostained pupae in brightfield (far left), fluorescence from these samples associated with lipophilic Nile red (staining predominantly fat body; center left) and hydrophilic Uranine (staining predominantly hemolymph; center right), and all three images merged together (far right). Within each representative photo, the specimen on the left is an unlabeled control and the one on the right depicts a representative labeled sample. All scale bars represent 1 mm. Ventral views of honey bee pupae (**a**–**d**), *V. destructor* protonymphs (**e**–**h**), deutonymphs (**i**–**l**), and foundresses (**m**–**p**). Each row of photos is a representative of at least 60 individuals from six independent experiments. The dorsal views of *V. destructor* mites are shown in Supplementary Fig. 1.

abdomen-feeding mites, although our study cannot specifically determine whether the exact feeding site influences which food is taken up.

**Food ingested by *Varroa destructor* mites from honey bee pupae**
Even though *V. destructor* feed primarily on the fat bodies of adult bees[33], a major part of the life cycle of *V. destructor* and *Tropilaelaps* spp. consists of the reproductive stage during which these bee mites feed on pupating honey bees, a host stage that differs from adults in physiology and morphology. Therefore, we investigated the hypothesis that *V. destructor* alternates between fat body and hemolymph as primary food source, depending on host life history and body composition. Our test used the methodology of Ramsey et al. of labeling host tissues with two fluorescent biostains, the hydrophilic Uranine and lipophilic Nile red, to distinguish the primarily hydrophilic hemolymph and lipophilic fat body in the mites after feeding[33]. We successfully produced fluorescently stained honey bee larvae in vitro that were able to develop normally into pupae. Fluorescent imaging verified that strong signals of Uranine and Nile red could be detected from the honey bee pupae (Fig. 3a–d). After parasitizing biostained pupae, foundress mites were observed to reproduce successfully and their offspring also grew normally, indicating that biostains allowed normal physiological processes to proceed.

Consistent Nile red and Uranine labeling was apparent in honey bee pupae (Fig. 3a–d). Quantification of the fluorescence signals from the abdomen of biostained pupa showed that the level of green fluorescence was about 10 times stronger than that of red fluorescence (Uranine/Nile red ratio = 10.19 ± 0.69). Nile red and Uranine signals could be detected in all mite samples. *V. destructor* feeding on fluorescently labeled pupae were disproportionally labeled with green compared to red, indicating that hemolymph instead of fat body is the primary food of mites when parasitizing honey bee pupae (Fig. 3e–p). Quantitatively, the Uranine/Nile red ratio in *V. destructor* protonymphs equals 309.17 ± 30.14, 666.02 ± 37.90 in deutonymphs, and 1689.81 ± 114.06 in foundresses (Fig. 4). Thus, all *Varroa* samples displayed significantly higher ratios than the honey bee pupae (overall model: $F_{(4,30)}$ = 1235.6, $p < 0.001$; all post hoc Dunnet T3 tests $p < 0.001$). Our additional study of fluorescence in *V. destructor* feces excluded a possible preferential excretion of fat body materials as explanation for the staining of *V. destructor* individuals because the feces also contained more Uranine with a Uranine/Nile red ratio = 161.46 ± 6.23, which is 16x more Uranine-enriched than the food source ($p < 0.001$), but less than the ratio in any *V. destructor* life history stage (all $p < 0.001$). Although the feces were likely derived from a mixture of mite stages[25], these combined results indicate that *V. destructor* of all ages consume more hemolymph than fat body when feeding on honey

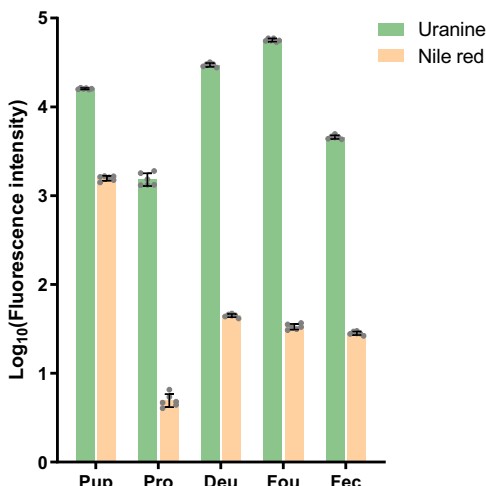

**Fig. 4 | Quantitative comparison of lipophilic Nile red and hydrophilic Uranine staining for _Varroa destructor_ feeding on stained honey bee pupae.** Nile red preferentially stains fat body and Uranine preferentially stains hemolymph. For honey bee pupa (Pup, _n_ = 6 replicates, 1 pupa/replicate) only the abdomen was collected for analysis. For protonymphs (Pro), deutonymphs (Deu), and foundresses (Fou) of _V. destructor_ mites, six biological replicates were prepared with each replicate containing three individual mites of the same development stage (_n_ = 6 replicates, 3 mites/replicate). For mite feces (Fec) samples, also six replicates were prepared, and in each replicate feces from three capsules were collected and pooled together (_n_ = 6 replicates, 3 capsules/replicate). Absolute fluorescence intensities were log-transformed. All data are presented as mean ± S.D. Source data are provided in the Source Data file.

bee pupae. Both stains are metabolically inert[46,47] and degradation of Nile red is therefore not a plausible alternative explanation either.

Our results are in strong contrast to staining results when _V. destructor_ parasitizes adults during the dispersal stage that show an enrichment of the lipophilic Nile red[33]. Nevertheless, we detected some low-level Nile red signals, which may be due to low-level fat body consumption or from lipophilic components of the hemolymph, such as lipophorins[48], hemocytes[49], or suspended fat body cells. Continuous consumption of the stained pupae presumably led to an accumulation and concentration of fluorescent signals that were consistently higher in deutonymphs than in protonymphs, while the Uranine/Nile red ratio also increased (Fig. 3e–l). The fluorescent staining of adult foundresses was not as strong (Fig. 3m–p), which could be due to a bias of the fluorescence measures as a consequence of the thick cuticle of adult _V. destructor_ despite our pretreatment to increase its transparency with peroxide. However, the stained in-situ photos were confirmed by our quantitation of the fluorescent signals of homogenized samples that did not rely on optical penetration of the cuticle. Thus, the results indicate that adult foundresses may not feed as much as developing protonymphs and deutonymphs relative to their body size and that they may be even more selectively feeding on hemolymph. Alternatively, they might be disproportionally excreting lipophilic substances. However, our analysis of the feces fluorescence makes this explanation quantitatively improbable.

**Analysis of the _Varroa destructor_ proteome**

To investigate which tissue serves as the main food source of _V. destructor_ when feeding on honey bee pupae, we sought to identify honey bee proteins in the mites that can readily be identified before complete digestion[35]. Similar proteomic profiles of ingested food and specific host tissues provide molecular evidence of feeding preferences. We performed this analysis comparatively on reproductive and dispersing _V. destructor_ to guard against potential biases that might favor the identification of proteins that are either derived from

the honey bee fat body or hemolymph. Preferential representation of hemolymph- or fat body-derived proteins could arise if one food type would be more easily digested than the other and therefore be less identifiable. In reproductive foundresses and dispersing _V. destructor_, 272 and 182 honey bee proteins were identified, with 105 and 15 proteins exclusive to reproductive and dispersing individuals, respectively (Fig. 5a). Overlap with a representative honey bee hemolymph proteome[50], was significantly higher ($\chi^2 = 10.3$, $p = 0.001$) for the proteins that were specific for foundresses (78/105 = 74.2%) than for proteins that were specific for dispersing mites (5/15 = 33.3%). The honey bee proteins that were identified from both dispersing and foundress mites contained 52.7% (88/167) hemolymph proteins (Fig. 5a). These pronounced differences in the proportion of hemolymph proteins among the three protein groups indicate that dispersing and foundress _V. destructor_ preferentially feed on different host tissues. However, these results cannot refute the possibility that foundresses also ingest some fat body material and dispersing _V. destructor_ also consume some hemolymph, as is also suggested by our biostaining results. Moreover, the considerable number of honey bee proteins found in both _V. destructor_ adult stages may be attributable to honey bee proteins that are not specific to hemolymph or fat body. Given the technical challenges to unambiguously identify proteins from small arthropod samples the overlap of 74.2% despite methodological differences is remarkable, but we cannot rule out that some minor hemolymph proteins were not identified by us or the previous study[50].

No raw data for a representative honey bee fat proteome profile was publicly available at the time of analysis to allow a complementary reverse of the analysis described above. However, some specific support comes from honey bee vitellogenin, a multifunctional egg yolk precursor protein with nutritional and antioxidant effects[51]. Vitellogenin was only identified in dispersing mites and while it might be used directly by _V. destructor_ for its reproduction[52] it is no longer detectable in foundresses that we sampled from purple-eyed honey bee pupae. At this stage, the foundresses have presumably exhausted any food reserves from their dispersal stage and rely solely on their developing host. Vitellogenin is primarily synthesized by the abdominal fat body of honey bees and is secreted into the hemolymph, peaking in young, nurse-age workers, but it is undetectable in pupal stages until the pharate adult stage, approximately 10 hours before eclosion[53–55]. Vitellogenin is therefore not an available food during the majority of reproductive stage of _V. destructor_. Compatible with these findings, the presence of honey bee vitellogenin in dispersing mites but not in foundresses supports the notion that the hemolymph and fat body are the primary food for _V. destructor_ during the reproduction and dispersal stages, respectively. In contrast, multiple _V. destructor_ vitellogenin proteins were identified in both, dispersing mites and foundresses.

Most honey bee-derived proteins that were present in both _V. destructor_ life history stages were more abundant in foundresses than in dispersing individuals (Fig. 5b). This suggests that host tissues are either more ingested or less digested during the foundress than the dispersing stage. These proteins include major royal jelly proteins (MRJPs) and hexamerins (Fig. 5c). MRJPs are secreted by the hypopharyngeal gland of nurse bees and fed to larvae as a major source of amino acids and have been repeatedly identified in the hemolymph of honey bee larvae and pupae and the fat body of adults[56–58]. However, MRJPs have not been found in adult hemolymph[50]. The identification of MRJPs in dispersing and reproductive mites and higher abundance in foundresses thus supports the view that _V. destructor_ switches from fat body-feeding on adults to hemolymph-feeding on pupae. Hexamerins are mainly synthesized by the larval fat body and secreted into the hemolymph for pupal development[59]. It is therefore present in both tissues and the high abundance of all four hexamerins (HEX 70a, HEX 70b, HEX 70c, and HEX 110) in _V. destructor_ foundresses thus cannot distinguish hemolymph- from fat body-feeding at this stage.

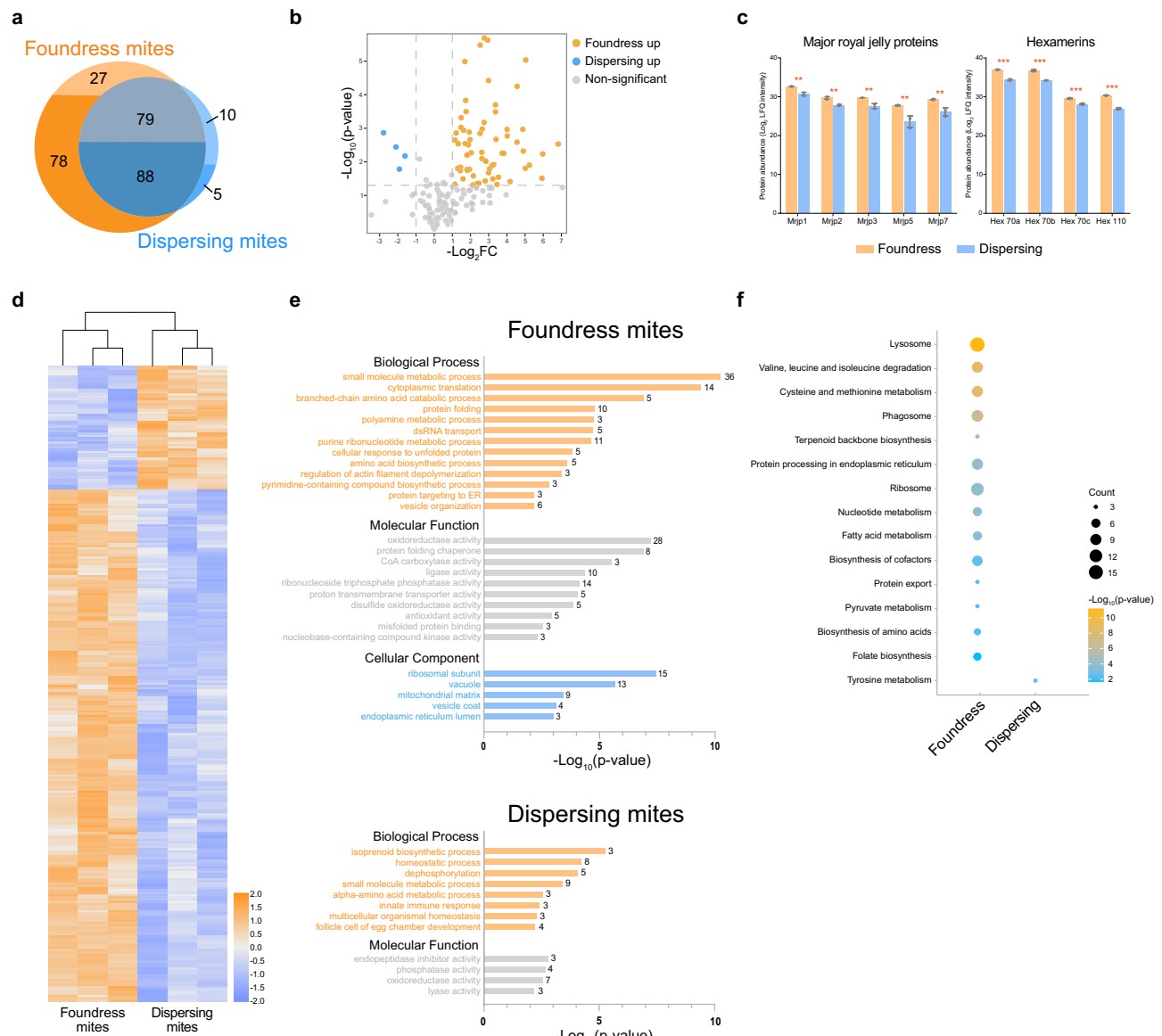

**Fig. 5 | Proteomic profiles of *Varroa destructor* foundress and dispersing mites.** **a** Most honey bee proteins identified in *V. destructor* were found in both stages, but almost 41.8% were uniquely identified in foundresses (parasitizing purple-eyed honey bee worker pupae) or dispersing mites (parasitizing adult nurse bees). The proportion of these proteins that had previously been identified in honey bee hemolymph (ref. 50) was highest in proteins unique to foundresses and lowest in proteins unique to dispersing mites with shared proteins intermediate, as indicated by the darker-colored sections. **b** Among the 167 honey bee proteins that overlapped between founding and dispersing mites, 69 were significantly different in abundance. The majority (65 proteins) was significantly more abundant in founding than in dispersing mites (yellow) and only four proteins showed the opposite result (blue). Statistical analyses were performed by two-sided t-tests with Benjamini Hochberg-correction. **c** Major royal jelly proteins and hexamerins from honey bees were consistently more abundant in foundresses than in dispersing mites ($n = 3$

biological replicates for each stage, approximately 50 mites/replicate). All data are presented as mean ± S.D. Significance of two-sided t-tests are indicated with ** and *** ($p < 0.01$ and $< 0.001$, respectively). Source data and the exact p-values are provided in the Source Data file. **d** Differentially expressed proteins of *V. destructor*'s own proteome (non-honey bee derived) clustered into a larger group of proteins up-regulated in foundresses and a smaller group up-regulated in dispersing mites. Statistical analyses were performed by two-sided t-tests with Benjamini Hochberg-corrected $p$ value < 0.05. **e** GO analysis of the upregulated *V. destructor* proteins in foundresses and dispersers revealed a variety of biological and molecular functions (two-sided hypergeometric test and Benjamini-Hochberg correction). **f** KEGG pathway analysis of the differentially expressed *V. destructor* proteins indicated a more active protein metabolism in foundresses compared to dispersing mites (two-sided hypergeometric test and Benjamini-Hochberg correction). Source data are provided in the Source Data file.

However, the quantitative difference to dispersing *V. destructor* demonstrates the utility of our proteome approach in general because the mite proteomes reflect the much higher expression of hexamerins in honey bee brood than in adults[60,61].

In addition to the honey bee proteins, our proteome comparison between foundresses and dispersing *V. destructor* identified 277 differentially expressed mite-endogenous proteins. These internally synthesized proteins do not indicate any particular food source but

their differential abundance informs our inferences about the physiological state of dispersing and foundress *V. destructor*. Relative to dispersing mites, 233 proteins were significantly upregulated in foundresses, while 54 proteins showed the opposite pattern (Fig. 5d). The proteins upregulated in foundresses were mainly involved in the biological processes of small molecule metabolic, cytoplasmic translation, and branched-chain amino acid catabolic process with the major molecular functions of oxidoreductase activity, protein folding

chaperone, and CoA carboxylase activity, and located in the cellular components of ribosomal subunit, vacuole, and mitochondrial matrix (Fig. 5e). In contrast, the upregulated proteins in dispersing mites were mainly enriched in the biological processes of isoprenoid biosynthetic process, homeostatic process, and dephosphorylation with the major molecular functions of endopeptidase inhibitor activity, phosphatase activity, and oxidoreductase activity, without any cellular component category being enriched (Fig. 5e). The proteins upregulated in foundresses were significantly enriched in 14 KEGG pathways, such as Lysosome, Valine, leucine and isoleucine degradation, Cysteine and methionine metabolism, and Phagosome, whereas dispersing mites upregulated proteins were only enriched in Tyrosine metabolism (Fig. 5f). The protein-protein interaction enrichment analysis using the differing *V. destructor* proteins identified Ribosome, Valine, leucine and isoleucine degradation, Protein processing in endoplasmic reticulum, and Phagosome as densely connected networks (Supplementary Fig. 2). These results reflect enhanced metabolism and protein synthesis in foundresses compared to dispersing *V. destructor*, which may be fueled by their diet of honey bee hemolymph.

Although the hemolymph might be less nutritious than the fat body[33], the profiling of host-derived proteins and the *V. destructor*-endogenous proteome suggest that the foundresses, feeding primarily on hemolymph, are sufficiently provisioned to sustain a very active protein metabolism. The potentially poorer quality of hemolymph may be offset by easy access to an abundance of hemolymph while parasitizing the pupae. Alternatively, left-over fat body nutrients from the previous dispersal stage might sustain the foundresses, although the quick death of *V. destructor* fed on hemolymph in an artificial environment[33] makes this explanation improbable. Thus, it is more likely under natural conditions that abundant hemolymph provides sufficient nutrients and hemolymph-feeding *V. destructor* certainly can survive for extended periods of time[62]. Thus, the above-reported feeding sites may simultaneously optimize access to large quantities of hemolymph and space use in the cell[26]. The fat body of honey bee pupae is more diffuse[40,63] than that of adult bees, in which limited areas of permeable intersegmental membranes coincide with a layer of fat body in the abdomen[63]. Our investigation of the fat body distribution during honey bee worker development confirms that fat body availability strongly differs between pupal and adult host stages. The abdominal fat body builds up slowly and gradually under the sternites and only is available in considerable amounts late in pupal development and adulthood (Supplementary Fig. 3). The feeding site may thus determine the primary food source for *V. destructor* and the most important selection may occur with regards to that site instead of the nutritional value of fat body versus hemolymph[33]. Under natural circumstances, the protected brood cell environment may allow *V. destructor* to thrive on hemolymph and dedicate most of their physiological capacity to reproduction, although this might not be sufficiently mimicked by artificial rearing conditions in which *V. destructor* suffer short life and little reproduction[33]. In contrast, the dispersal period involves active attachment to adult bees and evasion of defense behaviors, such as grooming. Therefore, dispersing *V. destructor* may require more energy for non-reproductive physiological processes, which is suggested by their relatively higher rate of heat production[64].

## Metabolomic profiling of *Varroa destructor*

To further investigate our hypothesis of stage-specific diets in *V. destructor*, we tested the prediction that the metabolomes of foundresses and dispersing females are distinct as a consequence of nutrition and food composition[65–68]. With our comparative metabolomic approach we also sought to characterize the impact of the alternative food sources on mite metabolic profiles across the life cycle.

The results indicated a high stability and reproducibility of our results because individual data points of the quality control and other sample groups were tightly clustered together while clearly separated from other groups (Fig. 6a). Both nymphal stages and the foundress stage were grouped together in a cluster analysis of the expression profiles of the differing metabolites in relation to the dispersing mites that were most divergent among the four groups (Fig. 6b). This pattern parallels the different food sources of dispersing *V. destructor* versus nymphs and foundresses and contrasts with an age or developmental pattern, which would predict foundresses and dispersing adults to cluster together. Numerous metabolic differences exist among the two adult stages (Fig. 6c). We cannot exclude that the environmental stress during the dispersal stage affects the mites' metabolism, switching to a stress-resistant "survival state" and the abundance of glycerophospholipids and lipids in dispersing mites (Fig. 6d) and corresponding KEGG pathways (Glycerophospholipid metabolism and Biosynthesis of unsaturated fatty acids, Fig. 6e) partially support this interpretation[69,70]. However, these results could also simply be a consequence of dispersing *V. destructor* feeding on the bees' fat body[33], which contains numerous lipid droplets[71] suggesting dietary causes.

The metabolites more abundant in foundress mites were significantly enriched in the pathways such as Aminoacyl-tRNA biosynthesis, Biosynthesis of amino acids, D-Amino acid metabolism, and ABC transporters, indicating a strongly enhanced metabolism of amino acids (Fig. 6e). Proximately, this can be explained because the insect hemolymph has a high concentration of amino acids[72]. Amino acids derived from pupal hemolymph can be directly used by mites for protein synthesis through translocation and activation, supported by the enriched ABC transporters and Aminoacyl-tRNA biosynthesis pathways. Ultimately, the foundresses need an upregulated protein metabolism to reproduce, a metabolic switch that can be activated by ToR signaling and must be accompanied by the availability of amino acids[73]. In addition, unlike the fat body, which mainly stores glycogen[39], hemolymph contains the disaccharide trehalose that can be more easily exploited to provide energy or synthesize other sugars. Correspondingly, the significantly enriched Citrate cycle and Biosynthesis of nucleotide sugars pathways imply an effect of hemolymph consumption on the carbohydrate metabolism of mites. Taken together, the different metabolic profiles of foundresses and dispersing mites support our hypothesis that *V. destructor* switch food sources from feeding mainly on hemolymph when parasitizing pupae to mainly feeding on fat body when parasitizing adult honey bees. This switch could be mandated by the different availability of these tissue in the different host life stages but a plausible adaptive explanation is the mites' different nutritional needs during the dispersal and reproductive stage[74].

Even though foundresses and nymphs feed on pupae in the capped cells, they also showed some marked differences among their metabolomes. Overall, more metabolites showed a higher abundance in protonymphs and deutonymphs than in foundresses. These were significantly enriched in a number of metabolic pathways, such as 2-Oxocarboxylic acid metabolism, Biosynthesis of amino acids, Glutathione metabolism, Aminoacyl-tRNA biosynthesis, Histidine metabolism, and beta-Alanine metabolism (Supplementary Fig. 4). Metabolites with higher abundance in foundresses were significantly enriched in Retinol metabolism and Neuroactive ligand-receptor interaction (Supplementary Fig. 4). Thus, the developmental stages show differences from the foundresses that are opposite to the dispersing mites, consistent with their fast growth and maturation. *V. destructor* is selected for an extremely fast development to be completed before the bee host emerges from the brood cell[75] and different utilization of the pupal hemolymph is possible but our data only provides a first speculative view into the developmental physiology of *V. destructor*.

## Comparative biostaining of *Tropilaelaps mercedesae* mites

The existing evidence suggests that *T. mercedesae* mites are exclusively parasitizing honey bee brood and are unable to use adult bees as a

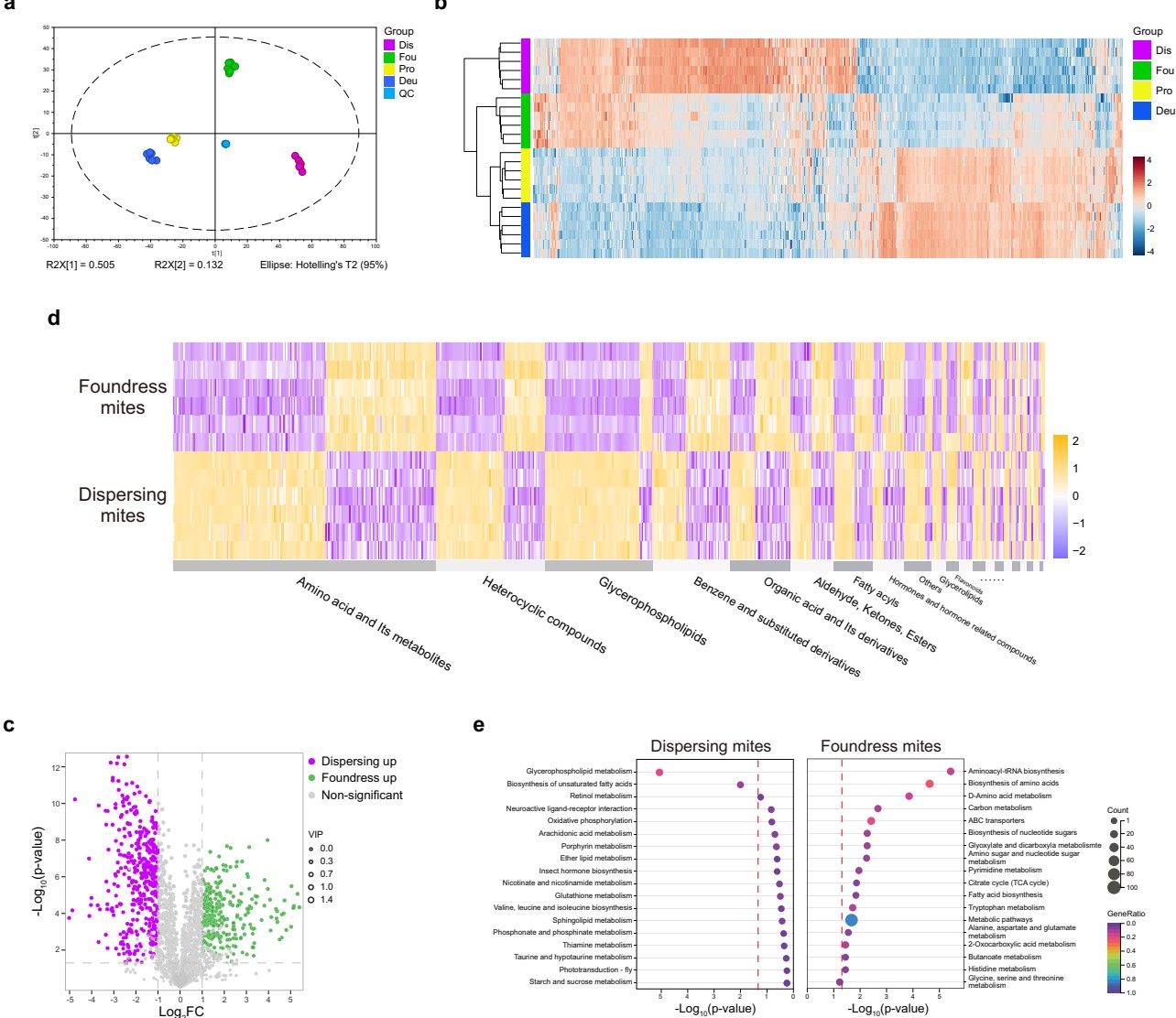

**Fig. 6 | Metabolomic profiling of protonymphs (Pro), deutonymphs (Deu), foundresses (Fou), and dispersing *V. destructor* mites (Dis) corroborate differences in food source between foundresses and dispersers. a** The PCA plot based on the complete metabolomics dataset displays a clear separation of all four life history stages with six replicates of each stage tightly clustered (indicated by different colors according to legend; QC = quality control). **b** Heatmap and clustering of metabolites that differed significantly in at least one of the pairwise comparisons. The adult foundress cluster is grouped with both nymphal stages (all raised on purple-eyed honey bee pupae) instead of the adult dispersal stage (sampled from adult nurse bees), indicating metabolic similarity according to diet instead of age. **c** Volcano plot to illustrate the 29.5% of all quantified metabolites that are more abundant in foundresses (green) and dispersing (purple) mites compared to the other life-history stage. Variable influence on projection (VIP) values were derived from orthogonal partial least squares-discriminant analysis. Statistical analyses were performed by two-sided t-tests. **d** Heatmap and classification of the metabolites that differ between foundresses and dispersing mites show a complex pattern with all metabolite classes containing some more and some less abundant metabolites in foundresses versus dispersing mites. **e** The KEGG pathways enrichment of upregulated metabolites in foundresses and dispersing mites indicates overall more protein-related metabolism and more fat-related metabolism respectively (two-sided hypergeometric tests). Source data are provided in the Source Data file.

food source[19,76]. *T. mercedesae* occasionally feed on brood even during their short dispersal stage and are thus completely dependent on honey bee brood[31].

On brood, *T. mercedesae* is a more versatile feeder than *V. destructor*: Feeding sites are mainly located on the abdomen but can also be found in diverse locations on the antennae, legs, thorax, and proboscis of the host[31]. This apparent flexibility of feeding location, coupled with the fact that *T. mercedesae* mites also show a strong bias towards Uranine-staining (Fig. 7), indicates that they also mainly consume hemolymph compared to fat body. Similar to *V. destructor*, *T. mercedesae* feeding on labeled honey bee pupae exhibited stronger signals of Uranine-labeled hemolymph than Nile red-labeled fat body,

although protonymphs, deutonymphs, and foundresses of *T. mercedesae* appeared not as different from each other as experimental groups of *V. destructor*. These visual in-situ results were confirmed by quantitative analyses (Fig. 8): *T. mercedesae* protonymphs had a Uranine/Nile red ratio of $205.05 \pm 19.91$, deutonymphs of $171.38 \pm 12.63$, and foundresses of $54.75 \pm 4.42$. Thus, sampled groups were significantly different (overall model: $F_{(4,30)} = 604.9$, $p < 0.001$; all post-hoc Dunnet T3 tests $p < 0.01$). However, these ratios in *T. mercedesae* varied less and exhibited a pattern that was inverse of what was observed in *V. destructor* with the highest ratio in protonymphs and the lowest in foundresses. Fluorescent staining of *V. destructor* was also higher than of *T. mercedesae* in absolute terms, ranging from 2.47x in

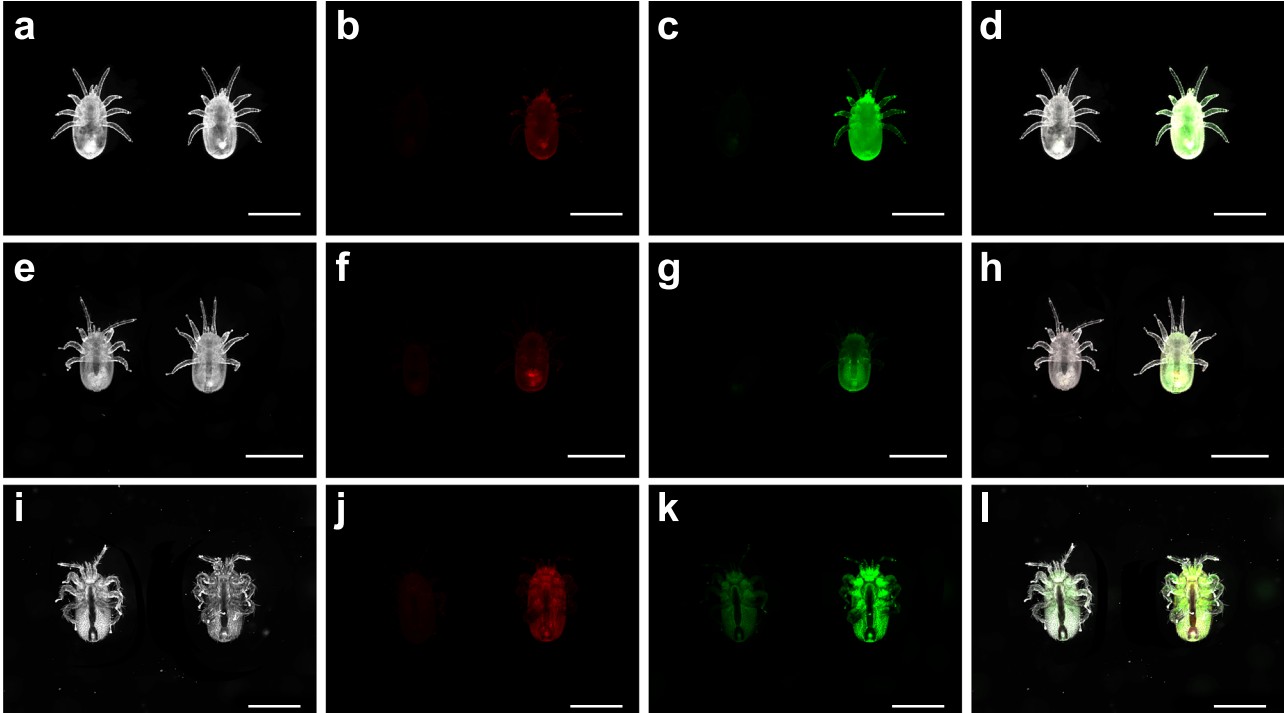

**Fig. 7 | *Tropilaelaps mercedesae* ingests large amounts of honey bee pupal hemolymph.** Photos show *T. mercedesae* mites fed on biostained pupae in brightfield, fluorescence from these samples associated with Uranine and Nile red, and all three images merged together. In each photo, the specimen on the left is the control and the one on the right is the treatment. All scale bars represent 1 mm.

Ventral views of protonymphs (**a–d**), deutonymphs (**e–h**), and foundresses (**i–l**) of *T. mercedesae* mites. Each row of photos is a representative of at least 60 individuals from six independent experiments. The dorsal views of *T. mercedesae* mites are shown in Supplementary Fig. 5.

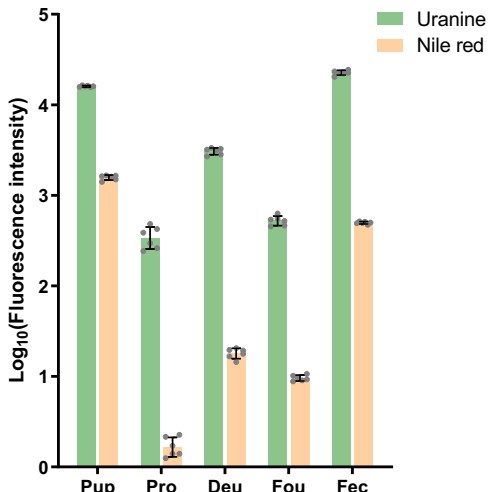

**Fig. 8 | Quantitative comparison of lipophilic Nile red and hydrophilic Uranine staining for *Tropilaelaps mercedesae* feeding honey bee pupae.** Nile red preferentially stains fat body and Uranine preferentially stains hemolymph. For honey bee pupa (Pup, *n* = 6 replicates, 1 pupa/replicate) only the abdomen was collected for analysis. For protonymphs (Pro), deutonymphs (Deu), and foundresses (Fou) of *T. mercedesae* mites, six biological replicates were prepared with each replicate containing three individual mites of the same development stage (*n* = 6 replicates, 3 mites/replicate). For mite feces (Fec) samples, also six replicates were prepared, and in each replicate feces from three capsules were collected and pooled together (*n* = 6 replicates, 3 capsules/replicate). Absolute fluorescence intensities were log-transformed. All data are presented as mean ± S.D. Source data are provided in the Source Data file.

protonymphs and 9.65x in deutonymphs to over 100x in the foundress stage. This considerable difference may be attributed to the faster metabolic rate of *Tropilaelaps* spp. which corresponds to the remarkable speed and maneuverability of these mites as adults[76]. In support of this explanation, particularly in terms of elevated excretion, the feces of *T. mercedesae* were found to be more dispersed and more intensely stained than *V. destructor* feces (Fig. 8). Thus, *T. mercedesae* might excrete the biostains more efficiently which could explain why stains seem to accumulate across developmental stages in *V. destructor* but not in *T. mercedesae*. Furthermore, the selectivity of feeding might increase with age in *V. destructor* in contrast to *T. mercedesae* but more detailed feeding studies are needed to clarify the observed patterns in detail. Orthology gene clusters that are specifically shared between the *T. mercedesae* and *Ixodes scapularis* genomes are enriched in renal tubule development functions, further corroborating the hypothesis of an increased excretory capacity of *T. mercedesae*[38].

**Comparative analysis of the *Tropilaelaps mercedesae* proteome**
The proteomic profiling of *T. mercedesae* mites revealed 258 honey bee-derived proteins, the majority of which were also found in *V. destructor* (Supplementary Fig. 6). Among these, 124 proteins overlapped with proteins found in both foundresses and dispersing *V. destructor*, 56 proteins overlapped with proteins that were only found in *V. destructor* foundresses and no overlap existed with honey bee proteins that were found specifically in dispersing *V. destructor* (Fisher's exact *p* < 0.0001). Among the 180 overlapping proteins, 113 proteins (62.8%) overlapped with a representative honey bee hemolymph proteome[50], while 55 of the 78 (70.5%) proteins that were uniquely found in *Tropilaelaps* overlapped. These data thus support our biostaining results that *T. mercedesae* feeds primarily on hemolymph, which has also been suggested previously[29]. The higher overlap in

honey bee-derived proteins between *T. mercedesae* and *V. destructor* foundresses compared to *T. mercedesae* and dispersing *V. destructor* consequently further supports our conclusion that *V. destructor* foundresses are primarily hemolymph feeders similar to *Tropilaelaps* spp[27].

Similar to many parasites, *Varroa destructor* is characterized by a life cycle with discrete stages to exploit different hosts. *Varroa* spp. are completely dependent on one honey bee host species but show distinct adaptations to parasitize adults during its dispersal stage and brood during its reproductive stage[36]. The dispersal, reproductive, and developmental stages may vary in nutritional needs and we show here that their life cycle involves a dietary alternation between hemolymph- and fat body-feeding. However, we argue here that food accessibility may also be important. It has been argued that *V. destructor* lacks crucial adaptations to feed on hemolymph[33]. However, their feeding apparatus is a suction pump that is capable of efficient fluid feeding[77] and the established feeding site that can be repeatedly exploited in the undisturbed brood cell may not require typical adaptations of other liquid-feeding ectoparasites that need to fill their nutritional needs in one short bout[78]. In contrast, the environmental conditions while feeding on adult honey bees during the dispersal stage are more challenging and the availability of hemolymph in adults is low, which may explain the primary consumption of fat body during that life history stage. Our findings are incompatible with the claim that *V. destructor* rely primarily on honey bee fat body[33] and hemolymph feeding appears to also be the predominant feeding mode the *T. mercedesae* mite, which completely relies on soft-skinned honey bee brood and cannot feed on adult bees[27,79].

The partitioning of life history into reproductive and non-reproductive stages is often driven by nutrient availability and the competing needs for dispersal and reproduction[80]. Our work exemplifies how these variables are interconnected in an ectoparasite that exploits the mobile adult stage of its host for dispersal and the stationary developmental stage for reproduction. The host stages thus match the parasite's requirements for locating a new host and subsequently exploiting it. The host life history stage also determines the available food and host defenses. The immobile brood is poorly defended and filled with abundant hemolymph, suitable for mite reproduction. In contrast, adult bees contain less hemolymph but more localized fat body tissues that coincide with sites on the hosts that offer some protection against behavioral defenses by the host[33]. Such host alternation between juvenile and adult stages of the same species is rare in ectoparasites and may be favored by the frequent close contacts of juveniles and adults in bee colonies and other social insects with overlapping generations. Such host species with many interacting individuals in close proximity and persistent colonies can lead to further reduction of the parasite's dispersal stage, as exemplified by the genus *Tropilaelaps*.

## Methods

### Observing mites with scanning electron microscope
Adult female bee mites were collected from capped brood cells. Mite samples were fixed using 2% paraformaldehyde and 2.5% glutaraldehyde at room temperature for 1 hour, followed by 3 washes with 0.1 M phosphate buffer (PB). Mite samples were subsequently fixed with 1% osmic acid at room temperature for 1 hour and washed 3 times with 0.1 M PB. The samples were dehydrated using an ascending series of ethanol (70%, 80%, 90%, 100%) and then dried with a $CO_2$ critical point dryer (EM CPD, Leica, Germany). Dried samples were gilded and observed by scanning electron microscopy (Quanta200, FEI, Czech).

### Detecting feeding sites of *Varroa destructor* on honey bee pupae
Eight honey bee (*A. mellifera*) colonies with natural infestations of *V. destructor* and *T. mercedesae* were maintained at the apiary of the Institute of Apicultural Research, Chinese Academy of Agricultural Sciences in Beijing. All colonies were queenright and managed using standard apicultural practices, except they were not treated to control mite populations.

To survey the feeding sites of *V. destructor* mites on honey bee pupae, frames containing capped worker pupae were removed from the hives between July and October 2022. Cells were opened appropriately 5 to 7 days after capping to search for pink-eyed to purple-eyed pupae with a white body[81] that were only infected with *V. destructor*. Feeding sites ($n = 1285$) were visualized by Trypan blue staining[82] under a Leica EZ4W microscope (Leica Microsystems, Germany) and the location of the feeding site was recorded.

### Breeding mites in the laboratory
To determine whether the diet of *V. destructor* and *T. mercedesae* mites during the reproductive stage consists predominantly of fat body or hemolymph, we followed the protocol of Ramsey et al. employing fluorescent biostains[33], except that larvae instead of adults were fed with Uranine and Nile red. Uranine is a hydrophilic fluorescent tracer[83] staining hemolymph, whereas the lipophilic Nile red[84,85] labels fat body[33]. In vitro honey bee larvae rearing was performed using a standard protocol[86]. Briefly, queens were caged on an empty comb to lay eggs. Newly hatched worker larvae were grafted into sterilized 48-well cell culture plates containing plastic queen cups and reared at 34.5 °C and 95% RH in a dark incubator. While a control group was fed a standard diet, Nile red and Uranine were added at the concentration of 0.25 mg/ml food during the final three days of feeding the biostained group of honey bee larvae.

Subsequently, the in vitro reared brood was used for the mite feeding experiments according to the previously established methods[87,88]. In short, foundress mites were collected directly from the sealed brood cells that had been capped for approximately 24 hours. The in vitro reared larvae were placed just before the start of pupation into size "0" gelatin capsules (7.34 mm diameter, Electron Microscopy Sciences, USA) with small custom-made ventilation holes. Foundress mites were individually transferred into a gelatin capsule containing one in vitro reared larva (one mite per capsule), and placed in an incubator at 34.5 °C, 75% RH in the dark. The protonymphs, deutonymphs, and adult mites were collected 1–2 days prior to worker emergence (grey wing pupae), and stored at −80 °C for further examination.

### Observing mites with fluorescence microscope
Before observation, adult *V. destructor* and *T. mercedesae* mites were submerged in 30% peroxide for 5 days to make the cuticle transparent. Nymphal mites were only rinsed with 70% ethanol to remove biostains that may be attached to their exterior, because their cuticles were sufficiently transparent without treatment. Then, pictures were taken with a Leica M205 FCA fluorescent stereomicroscope equipped with a Leica DFC7000T digital color camera (Leica Microsystems, Germany). The fluorescent signal of Nile red was acquired through an RFP filter set (541–551 nm excitation, LP 590 nm emission), and the Uranine signal was acquired through a GFP filter set (450–490 nm excitation, LP 500 nm emission).

Additionally, the distribution of fat bodies in the abdomen of developing worker pupae and adults were investigated. In vitro reared honey bee larvae were fluorescently stained as described above. Subsets of resulting biostained pupae were sampled every other day from the start of pupation until adult emergence. The abdominal sternite was cut out and briefly washed with PBS. Then, the inner surface of the abdominal sternite was photographed using a Leica M205 FCA fluorescent stereomicroscope equipped with a Leica DFC7000T digital color camera. Photos were captured and processed using LASX software (v2.02.15022).

### Fluorescence intensity assay
To quantify the amount of both biostains in the samples, Uranine and Nile red were extracted and measured in a fluorescence multiplate

reader (Tecan infinite 200 Pro, Tecan, Austria) using 96-well amber plates.

For the honey bee pupae, six individuals were prepared at the purple-eye stage for both the control group and the biostained group, with one pupal abdomen in each replicate. For *V. destructor* and *T. mercedesae* samples, six pooled replicates of three individuals each were prepared for each of the three developmental stages (proto-nymphs, deutonymphs, and foundresses) and the control and bios-tained groups. For mite feces samples, six replicates were prepared for both the control group and the biostained group, and in each replicate feces from three capsules were collected and pooled together.

Uranine and Nile red were extracted sequentially from samples with $H_2O$ and DMSO, respectively. Honey bee and mite samples were rinsed with 70% ethanol and air-dried first to remove biostains that may be attached to their exterior. Fecal samples were used without prior wash. Each sample was homogenized in 100 µl distilled water. The mixture was ultrasonicated for 5 min, followed by centrifugation at 12,000 g for 10 min. The supernatant was transferred to a new tube and the pellet was homogenized with 100 µl DMSO, mixed, and the supernatant collected as described above. The fluorescence intensity of Uranine was determined at an excitation-emission wavelength of 460 nm/515 nm, and the Nile red at 530 nm/635 nm.

### Collecting *Varroa destructor* samples for proteomic and meta-bolomic analyses

*V. destructor* mites in the reproductive stage were collected from capped brood cells containing purple-eyed pupae using a soft paint-brush and soft tweezers. Only foundress mites that were alive and accompanied by at least one protonymph and one deutonymph were sampled along with all protonymphs and deutonymphs.

For collecting dispersing mites, all capped brood frames of each colony (*n* = 8) were removed to prevent any reproductive mites from accidentally being released from emerging brood cells. After seven days, one open brood frame containing a mixture of young brood with nurse bees was taken from each colony and dispersing mites were collected from the abdomen of adult bees.

For each group (protonymphs, deutonymphs, foundresses, and dispersing mites), collected mites from all eight colonies were ran-domly allocated to one of six replicates of 20 mg each (corresponding to approximately 100 protonymphs or 50 individuals for all other life history stage) for the metabolomic analysis. Three additional repli-cates (50 mg for each replicate) of adult reproducing mites (foun-dresses) and dispersing mites were prepared for proteomic analysis. All samples were rinsed with PBS and air-dried to remove honey bee tissues that may have been attached to the cuticle, flash-frozen using liquid nitrogen, and stored at −80 °C for further processing.

### Proteomic profiling of foundress and dispersing *Varroa destructor* mites

**Protein extraction and digestion.** Total protein was extracted according to our previously described method with some modifica-tions. *V. destructor* mites (50 mg = approximately 125 individuals per replicate) were homogenized with lysis buffer (8 M urea, 2 M thiourea, 4% 3-[(3-cholamidopropyl) dimethylammonio]-1-propanesulfonate (CHAPS), 20 mM Tris-base, and 30 mM dithiothreitol (DTT)), and ultrasonicated on ice for 30 min. The supernatant was collected after centrifugation, and mixed with ice-cold acetone for protein precipitation[89]. Subsequently, the mixture was centrifuged twice at 12,000 g for 10 min at 4 °C, and the pellets were redissolved in lysis buffer. Protein concentration was determined using a Bradford assay and the general quality of extracted proteins was confirmed by SDS-PAGE with Coomassie Blue staining. Protein digestion, peptide desalting, and peptide quantification were performed as previously described[90]. In brief, mite protein samples (200 µg) were reduced with DTT (final concentration 10 mM) for 1 hour, then alkalized with iodoacetamide (final concentration 50 mM) for 1 hour in the dark. Thereafter, protein samples were digested at 37 °C overnight with sequencing grade trypsin (enzyme: protein (w/w) = 1:50). The digest was stopped by adding 1 µl of formic acid and desalted using C18 columns (Thermo Fisher Scientific). The desalted peptide samples were dried and dissolved in 0.1% formic acid in distilled water, then quantified using a Nanodrop 2000 spectrophotometer (Thermo Fisher Scientific) and stored at −80 °C for subsequent LC-MS/MS analysis.

**Proteomic profiling by LC-MS/MS.** LC-MS/MS analysis was per-formed on an Easy-nLC 1200 (Thermo Fisher Scientific) coupled Q-Exactive HF mass spectrometer (Thermo Fisher Scientific). The peptides were loaded onto a reverse phase trap column (Thermo Scientific Acclaim PepMap100, 2 cm long, 100 µm inner diameter, fil-led with 5 µm Aqua C18 beads) connected to the C18-reversed phase analytical column (Thermo Scientific Easy Column, 10 cm long, 75 µm inner diameter, filled with 3 µm Aqua C18 beads) in buffer A (0.1% Formic acid) and separated with a linear gradient of buffer B (80% acetonitrile and 0.1% Formic acid) at a flow rate of 300 nl/min. The following 90 min gradient was applied: from 3 to 8% buffer B in 2 min, from 8 to 23% buffer B in 63 min, from 20 to 40% buffer B in 12 min, from 40 to 100% buffer B in 3 min, and remaining at 100% buffer B for 10 min. The eluted peptides were injected into the mass spectrometer via a nano-ESI source (Thermo Fisher Scientific). The mass spectro-meter was operated in positive ion mode and all data were collected in a data-dependent mode with the following settings: scan range: m/z 300–1800; full scan resolution: 70000; AGC target: 3E6; MIT: 20 ms. For MS/MS mode, the following settings were used. Scan resolution: 17500; AGC target: 1E5; MIT: 60 ms; isolation window: 2 m/z; normal-ized collision energy: 27; loop count 10; dynamic exclusion: 30 s; dynamic exclusion with a repeated count: 1; charge exclusion: unas-signed, 1, 8, > 8; peptide match: preferred; exclude isotopes: on. The corresponding raw data were retrieved using Xcalibur software (v3.0, Thermo Fisher Scientific).

The extracted MS/MS spectra were searched against a protein database combined with *A. mellifera*, *V. destructor*, and all viruses known to infect *A. mellifera* and *V. destructor* (58,726 protein sequen-ces in total from NCBI) using MaxQuant software (v.1.6.17.0). These combined databases allowed us to distinguish proteins that were present in the samples because they were produced by *Varroa* from proteins that were taken up by feeding on honey bee hosts and not completely digested. The search parameters were as follows. MS1 match tolerance: 20 ppm for the first search and 6 ppm for the main search; MS2 tolerance: 20 ppm; enzyme: trypsin; allow non-specific cleavage at none end of the peptide; maximum missed cleavages per peptide: 2; fixed modification, Carbamidomethylation; variable mod-ifications: Oxidation and Acetylation (N-term); maximum allowed variable PTM per peptide: 3. A fusion target-decoy approach was used for the estimation of false discovery rate (FDR) and controlled at < 1.0% both at peptide and protein levels. Proteins were identified based on at least one unique peptide. For conservatively identifying honey bee proteins from our *V. destructor* samples, only proteins that were identified by MS/MS in at least 2/3 replicates in each sample group were included. Quantitative protein analyses to compare foundresses and dispersing mites were performed using label-free quantitation (LFQ) intensity (resulting from MaxQuant normalization) by Perseus software (v1.6.2.3). Briefly, reverse hits, proteins only identified by site, and contaminants were removed followed by filtering for proteins identified in four or more samples (Supplementary Data 1). Data was then Log2 transformed and missing values were not imputed. The two-sided t-tests were performed using SPSS (v20.0, IBM) and p-values were Benjamini Hochberg-corrected at 5% FDR.

**Proteomic data analysis.** Hierarchical clustering was performed using TBtools software (v1.1043)[91] based on average Euclidian distance.

Functional enrichment analyses, including Gene Ontology (GO) enrichment, Kyoto Encyclopedia of Genes and Genomes (KEGG) pathway enrichment, and protein-protein interaction enrichment analyses were performed using Metascape (v3.5.20230501)[92].

### Proteomic profiling of *Tropilaelaps mercedesae* mites

As *T. mercedesae* mites are rarely found on adult bees and do not have a typical dispersal stage, *T. mercedesae* were collected from capped brood cells containing purple-eyed pupae using a soft paintbrush and soft tweezers. About 600 adult *T. mercedesae* were collected from eight honey bee colonies and randomly allocated to one of three replicates, each containing 50 mg of mites. Samples were rinsed with PBS, air-dried, flash-frozen in liquid nitrogen, and stored at −80 °C until further processing. The proteomic analyses of *T. mercedesae* were performed as described above for *V. destructor*, except a protein database combined with *A. mellifera*, *T. mercedesae*, and all viruses known to infect *A. mellifera* (37,955 protein sequences in total from NCBI) was used (Supplementary Data 2).

### Metabolomic profiling of *Varroa destructor* mites

**Metabolite extraction from *V. destructor* mites.** To prevent degeneration, all procedures were carried out on ice, and all reagents for extraction were pre-cooled to −40 °C before use. Each sample pool was finely macerated in 400 µl 80% methanol. The mixture was ultra-sonicated for 15 min, followed by centrifuge at 12,000 g for 10 min at 4 °C. The supernatant (300 µl) was collected and placed in −20 °C for 30 min, then centrifuged again at 12,000 g for 3 min at 4 °C. A 200 µl aliquot of the supernatant of each sample was used for LC-MS analysis. Two blank control samples were prepared with 80% methanol alone and were subjected to the same method above. Additional quality control (QC) samples were prepared by mixing equal volumes from each of the samples.

**Metabolomic profiling by HPLC-Q-TOF-MS/MS.** MS/MS analyses were performed using a high-performance liquid chromatography (HPLC, LC20, Shimadzu, Japan) coupled to a TripleTOF 6600 mass spectrometer (AB SCIEX, USA).

For LC separation, the ACQUITY UPLC HSS T3 C18 (10 mm long, 2.1 µm inner diameter, filled with 1.8 µm Aqua C18 beads; Waters, USA) was used. The column temperature was 40 °C, the flow rate was 0.4 ml/min and the injection volume was 2 µl. A gradient elution program was run with mobile phase A (0.1% formic acid in water), and mobile phase B (0.1% formic acid in acetonitrile) as follows: 0 ~ 11 min, 95% A–10% A; 11 ~ 12 min, 10% A–10% A; 12 ~ 12.1 min, 10% A–95% A; 12.1 ~ 14 min, 95% A–95% A. The processing order of mite samples was randomized to avoid any systematic bias. QC samples were inserted into the sample queue (after every 15 samples) to monitor and evaluate the stability of the system and the reliability of the data.

The Triple TOF 6600 mass spectrometer (AB SCIEX, USA) was applied for mass spectrometry analysis. Electrospray ionization (ESI) positive and negative modes were used for detection. The ESI source conditions were as follows. Ion source gas1: 50 psi; ion source gas2: 60 psi; curtain gas: 25 psi, source temperature: 500 °C; declustering potential: 60 V ( + )/−60 V (-); and ion spray voltage: 5500 V ( + )/−4500 V (-). The data acquisition was performed with the information-dependent acquisition (IDA) mode. The TOF MS scan parameters were set as follows. Mass range: 50–1000 Da; accumulation time: 200 ms; and dynamic background subtraction enabled. The product ion scan parameters were set as follows. Mass range: 50–1000 Da; accumulation time: 50 ms; collision energy: 30 V ( + )/−30 V (-); collision energy spread: 15; resolution: UNIT; charge state: 1 to 1; intensity: 100 cps; exclude isotopes within 4 Da; mass tolerance: 50 ppm; maximum number of candidate ions to monitor per cycle.

**Metabolomic data analysis.** The raw data file generated by LC-MS was converted into mzML format by ProteoWizard software (v.3.0.4416). Peak extraction, peak alignment, and retention time correction were performed by the XCMS program. The support vector regression (SVR) method was used to correct the peak area, and the peaks with a deletion rate of > 50% in each group were filtered out. Subsequently, metabolic identification information was obtained by searching first against an in-house database that contains chemical standards and a manually curated compound list based on accurate mass (m/z, ± 25 ppm), retention time, and spectral patterns; then, the generated MS1/MS2 pairs were searched in the public databases: HMDB (http://www.hmdb.ca/), MoNA (http://mona.fiehnlab.ucdavis.edu/), and MassBank (http://www.massbank.jp/). In addition, the metabolic peaks with MS/MS spectra that were not matched in public databases were analyzed by MetDNA (http://metdna.zhulab.cn/)[93]. Results are available as Supplementary Data 3.

Unsupervised PCA (principal component analysis) and supervised orthonormal partial least-squares discriminant analysis (OPLS-DA) were performed with the software package SIMCA (v14.1, Umetrics AB, Sweden). Differential metabolomes were compared between the following pairs: Dispersing mites vs foundresses, foundresses vs deuto-nymphs, foundresses vs protonymphs, and deutonymphs vs protonymphs. The significantly different metabolites between experimental groups were determined by variable influence on projection (VIP ≥ 1) values derived from the OPLS-DA result, p-value (two-sided t-test, < 0.05), and fold change (FC ≥ 2). Hierarchical clustering was performed using TBtools software based on average Euclidian distance. Identified metabolites were annotated using the KEGG compound database (https://www.kegg.jp/kegg/compound/) and annotated metabolites were then mapped to the KEGG pathway database (https://www.kegg.jp/kegg/pathway.html). Significantly enriched pathways were identified with two-sided hypergeometric tests ($p < 0.05$) for each list of metabolites.

### Reporting summary

Further information on research design is available in the Nature Portfolio Reporting Summary linked to this article.

## Data availability

The proteomic data generated in this study have been deposited in the Proteome Xchange Consortium with the dataset identifier PXD047191 (https://www.iprox.cn//page/SCV017.html?query=IPX0005580000). All data supporting the findings of this study are available in the manuscript or supplementary information. Source data are provided with this paper.

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

## Acknowledgements

We thank all of our team members for their support. This study was funded by the Modern Agro-Industry Technology Research System (Grant No. CARS-44-KXJ-6) and the Agricultural Science and Technology Innovation Program (Grant No. CAAS-ASTIP-2022-IAR) to S.X.. Additional support was provided to O.R. from PAm a Bayer's "Health Hives" initiative and the Natural Sciences and Engineering Research Council of Canada (RGPIN-2022-03629).

## Author contributions

B.H., J.W., and S.X. designed the research; Q.W., J.W., B.H., F.L., and L.C. performed the research; B.H., O.R., and J.W. analyzed the data; S.X. provided resources; B.H. and O.R. interpreted the results and wrote the paper; and O.R. and S.X. edited the manuscript.

## Competing interests

The authors declare no competing interests.
