## [Peer Review File · Nature Communications]

Life-history stage determines the diet of ectoparasitic mites on their honey bee hostsREVIEWER COMMENTS

Reviewer #1 (Remarks to the Author):

This manuscript describes identifying the site of feeding in the case of Varroa and Tropilaelaps mites both parasitizing on honeybees. Sophisticated biochemical techniques were used to determine the location where these parasites sting the bee larvae and suck their food. Different food sources (hemolymph and fat body) for the Varroa mites were identified, which puts earlier publications on this subject into perspective.

Although the data now available clarifies previously divergent opinions, the novelty of this report is limited. Such data is nice to know, but the results obtained from it are not very surprising for the expert and do not really open new areas of knowledge. These experiments are very laborious and creditable, but this diligence should desirably lead in direct consequence to new practical knowledge (e.g. mite treatment) and/or new important physiological insights. These data support the conclusions. An additional experimental approach would greatly enhance the value of this manuscript: the joint exposure of both mites on one host with newly expected data sets. Especially influences during co-cultivation on the reproduction and dietary situation rate would be an innovative new approach. This topic will increase the scientific significance of the manuscript for the interested reader.

Minor comments:

line 4 - three authors as first authors who contributed equally are highly unusual
line 99 that Tropilaelaps mites pose a greater health risk to the honey bee is not proven
lines 525 and 535 - number of mites extracted should be given

Reviewer #2 (Remarks to the Author):

I have reviewed with great interest this manuscript on mite feeding physiology, it is a great piece of work, with a lot of experimental efforts. This study tackles an important yet unresolved question on mite physiology in the honey bee, with a partial comparative approach on varroa and tropilaelaps. Using physiology and omics it shows that varroa are most likely to feed primarily on haemolymph during its reproductive phase, while it preferentially eats from fat bodies during dispersal. I have several comments that would prevent publication of the manuscript in its present form.

First, it is presented as a comparative approach between Varroa and Tropilaelaps, but the comparison is not maintained throughout the results so I do not think that the title and claims fully match the work. In fact Tropilaelaps is only studied in the staining experiments, targeted on the reproductive phase. So unless I am wrong no data is presented for this mite on the dispersal phase. There is also no data on the proteomics and metabolomics.

So I would center the study around varroa mainly, and maybe have a little spotty comparison with Tropilaelaps.

I also find a real shame that no staining approach has been done on the dispersal phase. In the title you explicitly mention the "life-history stage" approach, but why doing it only for the proteomics and metabolomics?

Detailed comments:

L39: Varroa destructor should be in italics

Introduction: it reads very well and is well documented.

L.95: I believe the the dispersal phase can be (much) longer than seven days. I had 4-6 days long as an average, but up to several months in Northern climates.

L.98: I would state both reproductive rates to highlight the difference more clearly.

I would rearrange the first part of the results, because at the moment it is not very clear what the

focus is, and why (varroa only, varroa and tropilaelaps). You will need to justify more clearly why the first part is done on both, and the second and third parts only on varroa. At the moment it is confusing.

I did not understand what the input is of the first experiment (Feeding sites of Vd on honey bee pupae) for the question. It needs to be better justified, or removed.

I would then make a comparative section on the food ingested by both mites while in capped brood (Fig 3 and Figure 5). Then I would develop Fig 4, which is a nice comparative conclusion. To make the story complete, I miss the staining done for mites in the dispersal phase. If for this part you only base your conclusions on the comparison with the literature (Ramsey et al) it should be clearly stated. For me staining of dispersing varroa and tropilaelaps need to be included.

L250-252: it is such a shame not to have the equivalent of Fig 2 for tropilaelaps.

L 287 and M&M section: the fact that you have chosen mites from purple eyed pupae is a big issue to me, and probably biases the results a lot (it has been shown previously that mite transcriptomics varies a lot along the bee capped development). I understand the choice to sample juveniles (that do not bring that much to the story - I would place these aspects as supplementary figures), but to sample a reproductive mite I would have targeted anything from prenupa to white eyed pupa. I believe that at purple eyed the foundress is not really reproducing anymore sensu stricto. So you probably miss the key window.

L.480-486: can you give insights into the success rate? Survival? Mite fertility? Fecundity? To show that the in vitro system really mimics what happens in the hive.

L.521-524: in your introduction you state that the dispersal phase is at maximum 7 days. So I believe that by leaving mites for 7 days without brood, you may force the system. I would have picked 4-5 days.

Reviewer #3 (Remarks to the Author):

This study throws a new and much needed look at the question of which host tissue do highly damaging ectoparasite mites feed on when infesting honey bees, the major managed pollinator. The authors follow up on a work that led to a paradigm shift, but which remained controversial and not yet accepted widely in the field. This new study aims at using more direct methods (proteomics and metabolomics) to determine the diet of the mite *Varroa destructor*. It also replicated the methods used in the previous work, but on another life history stage of mites and thus provides a more complete picture of the nutritional adaptation of the parasite to the host. The authors conclude that the mites shift diet from using principally fat bodies when infesting adult hosts to hemolymph when infesting honey bee brood during their reproduction phase. They also performed some of the experiment on another ectoparasitic mite, which represent a threat to beekeeping worldwide and discuss the differences in diet adaptation of these two mite species.

A main issue I have with the way the manuscript is written is that the introduction does not set the stage of the study properly so that one wonders why some experiments were done and how they support the main aim. Clear hypotheses and expectations for each experiment are missing. Also, most of the data does not clearly support the conclusions, at least not the way it is presented, because of some apparent circular reasoning and poor connection between the results of the various experiments. An important piece of information missing is the structure of fat bodies in the pupae used for the biostaining experiment. This knowledge is fundamental to assess whether the difference in diet proposed is likely determined (as the title suggests) by host life history stage. Overall, the manuscript does not convince the reader and this could be due to the poor support of the conclusion by the data

or by a suboptimal presentation of the data.

Major comments

Some of the results presented go beyond the rationale of the study presented in the introduction. The analysis of feces comes out of the blue in the results and it is not clear how studying the omics of the immature mites contributes to the main aim: the determination of diet when feeding on brood.

Together with the lack of hypothesis statement and expectations making it difficult to determine the strength of the result interpretation, this gives a confuse impression to the reader who does not end the manuscript convinced of the conclusions proposed. The logic of several arguments is not clear, with some appearing circular. This should be improved to better make use of the results presented and increase the impact of the paper.

The data presented for *Tropilaelaps* derives from the same methodology as used in the previous work on *V. destructor* which was judged as insufficient (L117-119) and prompted the new study.

Tropilaelaps were not subjected to proteomics and metabolomics to further support the argument of a haemolymph diet. The evidence for *Tropilaelaps* thus appears weak and does not bring the question of the diet of these ectoparasites substantially further. Furthermore, the feeding sites of *Tropilaelaps* were not determined. The *Tropilaelaps* part of the work could be the topic of a separate paper without weakening the study much.

L138-140: I do not agree with this conclusion because the current hypothesis that *Tropilaelaps* feeds on brood only is because of anatomical limitations of their mouth pieces. Your argument requires hypothesising a reduction in mouth piece strength following the reduction to a pure hemolymph diet on the brood.

L119-124, 342-346: whether parasite diet is constrained by external anatomy of the host or a choice of the mite is indeed a very relevant and has not been sufficiently addressed. However, there are not enough details as to when the tissue structure of adults (fat bodies lining the cuticula) is acquired during pupal development. Here you only mentioned a diffuse pattern in metamorphosing individuals and not in the pupal stages tested. This information is fundamental to determine the relevance of the pupal stage you selected for your dye experiments (which by the way is not mentioned in section 4.4) and for the interpretation of your results. If fat bodies are already lining up the cuticula when reproductive mites are feeding on the hemolymph of these pupae, as proposed, then mites can choose their diet and are not constrained by host internal or external anatomy. This is especially important because you show that the feeding site on pupae is virtually the same as on adults.

On this note, you also should better discuss the incompatibility of your results with Ramsey's et al feeding experiments, which showed that mites need access to fat bodies to better survive and reproduce, while you show they do not have such access.

L134: I argue that your experimental design does not allow you to suggest that hemolymph diet is more beneficial to mite offspring. You have not manipulated their diet to show this.

L278-283: these proteomics results are not discussed and because of the lack of hypothesis or expectations presented for these analyses in the introduction, I have no idea how they contribute to the main aim.

L331-333: what allows you to determine that the differences observed with this proteome comparison supports the hemolymph feeding by foundresses? To me this sounds like an assumption and circular reasoning, not a piece of evidence. Protein metabolism is determined by the physiological needs (here oogenesis probably), not by the type of diet as long as proteins are present. These proteins could very well come from the fat bodies.

L334-336: same comment, you assume hemolymph feeding, the proteomics data does not show it!

L364: the strength of the metabolomics approach in identifying the diet type of each mite life history stage is weakened by the mere 'usual reflection' of diet difference in the metabolome. Excluding that the metabolome cannot be different on the same food source would be more powerful to support your conclusion. However, Fig 7C showing different patterns for foundress mites and its offspring, show a different metabolome on the same diet is actually possible.

Other comments

Title: The title could be improved to give a more representative idea of the work. Replacing 'dietary

alternation' by 'diet' would improve readability. However, because you have not shown that the mites do not have the ability to choose what they feed from, you cannot write that the stage determines the diet (this is only a hypothesis at this stage, see your L171-172). You can only state that the diet differs between the stages. See major comment on the structure of fat bodies in pupae and on anatomic constraints to diet.

I suggest referring to the species by their full scientific names throughout the manuscript. There are several *Varroa* and *Tropilaelaps* species in these genera, but only one of each has been experimented on here. When several species are meant, the genus name followed by 'spp.' is required.

Abstract:

L16: should probably be 'genera' instead of 'genus'

L25: the manuscript does not present proteomic data for *Tropilaelaps*, this sentence is thus misleading.

L32-33: your experimental design does not allow you to show the adaptation itself, a more careful formulation is required.

L38: I am not convinced that the issue is resolved, especially for *Tropilaelaps*.

L39: *V. destructor* should be in italics

L33, 40, 43, 365: can one speak of specialization or of evolved a strategy if the diet choice is constrained by the host internal anatomy? This possibility is not excluded by your work. See major comment on anatomical constraints.

Introduction:

L61: 'Eastern' instead of 'Easter'

L78: the female is not gravid (ie bearing oocytes) yet when she enters the cell, see also your text L91.

L76-88: contain details not required for the understanding of the rationale of this study and should be omitted. Eg initial egg developing into a male etc.

L83: 'rapidly' is too vague, please be precise, but see previous comment.

L86: only male and female offspring will mate

L104: it is unclear to me how maturation links to reproduction mentioned earlier

L101-110: at this point it is difficult to determine why this information is given. It is useful for the study? Will this information be used again? If yes, this should be made clearer. Clarifying the rationale, hypotheses tested and expectations of the study will help making sense of this information.

Alternatively, please consider giving this information in the discussion as you put your results in perspective.

L117-119: Please specify what information is lacking to show more directly which diet is used by the mites and explicitly mention how the study fills this gap.

L125 and following: I suggest removing the results from this section.

L135: please clarify which stage is meant here

L150: at this stage, we do not know if the nutrition is constrained by the feeding site, see your L171-172. This kind of information relating to the rationale of the study should be presented in the introduction, see major comments.

L154: a confirmation requires a citation for the work confirmed

L157-159: can this not be excluded with results of citations 42 and 43?

L160: please provide figures with the same level of precision throughout (one decimal) for consistency.

L164-166: why would this be useful? Please expand the discussion and do not let the reader guess the importance of this information.

L173: '...mites feeding on brood'. *Varroa* mites do not feed on uncapped brood, so no need to mention capping here.

L174-182: contains superfluous repetitions of the introduction and hypotheses which should already be given in the introduction.

L178: I disagree, they feed on the same host, but at a physiologically different stage.

L186: not only larvae, pupae too.

L187: how does this validate biostain quantification? The logical link is unclear to me.

L190: pupae do not feed. I suggest precisising which tissues are stained by which dye so that the reader does not need to consult the text to find out.

L197: the consistency of the staining is not shown in Fig 3 (only one representative individual chosen) and repeats L184-185.

Fig.4: the readability of this figure would be increased by separating the two mite species. Here too reminding, in the caption, which dye labels which tissue would help the reader. Please also mention the pupal stage on which the mites fed.

L228-233: the visual assessment of fluorescence intensity is not reliable and the quantitative comparison should be used here. The latter does not show a weaker intensity of foundresses compared to deutonymphs. Although no statistics are available to quantify this difference or lack thereof.

L238: I am not convinced by this argument because, as you mention L208, feces are a mixture originating from all stages.

L242: is an interpretation of the results and rather belongs to the main text

L248-253: belong to the introduction and should support the rationale for the study.

L254: indicates

L258-259: the results just mentioned refer to fig 4 which shows the results of the quantitative analyses, and not the visual results mentioned here.

L261-262: this is hard to see as the figure shows the values of these parameters and not their ratio. Could the figure only show ratios? Or add another panel with the ratios?

L263-271: interesting but off topic regarding the main aim of the study in my opinion.

L273-278: again this should be part of the introduction to support the rationale of the study. In addition, It is not clear how you identify proteins in Varroa that can readily be identified before complete digestion. Is this ahead of the analyses or post hoc? I do not understand the logic of how the comparison between dispersing and reproductive mites can guard against methodological biases. Which biases by the way?

L285-288: this inconsistency deserves an explanation!

L295: the use of 'respectively' in this sentence should be revised. Respectively to what?

L298-299: what should be concluded from this difference? Please discuss.

L303-306: could this difference not be due to a higher consumption of the founding mites to sustain oogenesis? Your hypothesis implies that dispersing mites store MRJPs acquired during the previous reproductive cycle and do not metabolise it. Is this possible?

Fig 6 would be more reader friendly with the mite type spelled out instead of abbreviated. Most of the text is of too small size for easy reading. Please mention the host developmental stage from which foundresses were collected in the caption.

Fig 6C: this is expression data, gene expression? Not protein quantity? Is this the correct terminology here and in the text? Against what was the data normalized?

L309-311: sorry, I do not follow your logic here. Something may be missing to make sense of this.

L340: the reader expects a more detailed explanation for this inconsistency!
A new paragraph should be started at the end of this sentence. And a better transition is needed to link the resulting paragraphs.

L346-351: this appears to be off-topic. I do not see any logical link with the previous sentences and the aim of the study.

L363-367: again these are details I would expect in the introduction associated with an hypothesis tested and expectations to support the use of either diet type.

Fig 7: Please mention host stage on which the foundress and offspring were collected. Most text is too small. A) The meaning of QC is not given.

L415-427: this appears off-topic and does not contribute to the main aim of the study in an obvious manner.

L425: emerges from

L439: a few words explaining why this is so would be desirable, instead of just citing a study.

L439-442: What do you mean with 'selective environment'? I do not follow the logic of this argument. What does this situation have to do with the source of their diet?

Methods

L485: please mention at which stage exactly. In addition, knowing the structure of fat bodies at this stage will help supporting your claims regarding their relation to mite diet, see main comments.

L517: omit 'directly'

Reviewer #4 (Remarks to the Author):

I appreciated that the Authors analyzed Varroa feces during their biostaining experiment/analysis. Compared to the study by Ramsey, et al. (2019), which did not present analysis of feces, the current study considers the rapid food processing of Varroa. Later (line 270 -271) the Authors cite research suggesting rapid food processing in Tropilaelaps. Are there similar data available for Varroa?

The Authors had the foresight to preempt Reviewers' critiques of the interpretation of their results. In several points (lines 158, 166, 171, for example), the authors place caveats on their results. This may have been intentional, but if not, doing so should help speed up the review process and makes it more difficult for Reviewers to offer criticisms (this is a good thing!).

The results from the metabolomic profiling analysis (as presented in Figure 7) were very convincing for the differences in nutrient sources for reproductive and dispersal stage Varroa mites.

Lines 19 – 20: The Authors should specify Varroa here...since their series of experiments, as reported in the manuscript, were focused on Varroa, and not Tropilaelaps mites. For example, there was no mention of proteomic analyses done for dispersal stage Tropilaelaps mites.

Lines 89 – 92: It seems that a citation(s) is needed here. Was all the information for Tropilaelaps mites reported here directly researched by those cited in (27)?

Lines 129, 130: Here the Authors mention mites are feeding on larvae. Although this may be true, later in the manuscript (e.g., line 137) they switch to using 'brood', and conduct the research reported in the study using specifically, honey bee pupae. Later, on line 186, 'larvae' is used, but in the figure just below this line, images of pupae are shown. The switch between using this-or-that word may be intentional, but it reads awkwardly. A related point to this, lines 480... gives methods for rearing hosts for Varroa foundresses, but I was confused at what stage (instar) the larvae are placed in the capsules. Can the Authors provide more details here?

Lines 152 – 154: Results from vital staining wound analysis are given here for Varroa. Given that the Authors had presented introductory information for Tropilaelaps feeding site preferences (line 81), it would be nice to see here a concise comparison between results for Varroa and those previously reported for Tropilaelaps. Also, how does Varroa feeding site preference compare between honey bee pupae and adult hosts? What can be said about this?

Line 201: 'labeled bees'...do you mean 'brood' or 'larvae'?

Line 214 (Figure 4): Was it possible to present fluorescence intensity of samples on a weight basis? If possible, how would doing so change the results presented in this figure?

Line 248: It would have been great to have tested this hypothesis by conducting proteomic/metabolomic analysis for dispersal stage Tropilaelaps...maybe the Authors are saving this for a subsequent paper?

Lines 257: From Figure 4, I am not sure this statement is accurate. The different stages of

Tropilaelaps look as different as those of Varroa.

Lines 312 – 315: Were the numbers reported here for those significant after correction for multiple comparisons? Same question for lines 355 – 357 (Figure 6 caption).

Line 512: What stage were the brood on these open brood frames? Please give this information. Could Varroa have fed on these larvae during the seven days?

Line 784: Check this citation. I think it should be S.D, Ramsey?

I believe the captions for Figures S1 and S2 should be more elaborate to include information for which biostain was used. They should read more like the captions given for Figures 3 and 5.

Figure S2: Why were representative images of Tropilaelaps feces not shown?

Dear Editors,

Thank you for giving us the opportunity to revise our manuscript NCOMMS-23-20145 and provide additional results. We would like to thank all reviewers for the time and detailed comments which we have addressed (in blue font) below. Due to the number of changes in the text, we have provided one manuscript file with tracked changes and one “clean” manuscript without tracked changes.

Sincerely, Olav Rueppell (for all co-authors)

Reviewer #1 (Remarks to the Author):

This manuscript describes identifying the site of feeding in the case of *Varroa* and *Tropilaelaps* mites both parasitizing on honeybees. Sophisticated biochemical techniques were used to determine the location where these parasites sting the bee larvae and suck their food. Different food sources (hemolymph and fat body) for the *Varroa* mites were identified, which puts earlier publications on this subject into perspective.

Although the data now available clarifies previously divergent opinions, the novelty of this report is limited. Such data is nice to know, but the results obtained from it are not very surprising for the expert and do not really open new areas of knowledge. These experiments are very laborious and creditable, but this diligence should desirably lead in direct consequence to new practical knowledge (e.g. mite treatment) and/or new important physiological insights. These data support the conclusions. An additional experimental approach would greatly enhance the value of this manuscript: the joint exposure of both mites on one host with newly expected data sets. Especially influences during co-cultivation on the reproduction and dietary situation rate would be an innovative new approach. This topic will increase the scientific significance of the manuscript for the interested reader.

REPLY: We thank the reviewer for confirming the technical thoroughness of our study. Our study was initially motivated by the paradigm-shifting impact of the publication by Ramsey et al. (2019), which we considered to be widely misleading the general public and scientific community (even though some experts shared out concerns from the beginning). We consider correcting a widely-cited and publicly discussed study in itself as significant. Moreover, the insights from our study may have implications for the development or modification of practical mite treatment. This is not only true for *Varroa*, but also for *Tropilaelaps*, an understudied but severe threat to the beekeeping industry. Finally, the discovery that ectoparasites switch their host-feeding behavior in response of their hosts' life-history stage is the first of its kind to our knowledge. This is a significant scientific discovery that is probably much more widely applicable but has not been investigated much, in contrast to textbook examples of host switching during the complex life cycles of many parasites. The suggestion to study co-infection of *Varroa* and *Tropilaelaps* is an excellent idea, which we are currently seeking funding for, but that is an independent topic that needs to be dealt with in a separate study.

Minor comments:

line 4 - three authors as first authors who contributed equally are highly unusual

REPLY: We would like to allocate authorship fairly and equally without cutting junior scientists out of the credit they deserve.

line 99 that *Tropilaelaps* mites pose a greater health risk to the honey bee is not proven

REPLY: We agree that it is not proven, although the claim can be found in the recent literature (Chantawannakul et al. 2018), and have therefore cautioned this statement with “potentially” and included the literature reference.

lines 525 and 535 - number of mites extracted should be given

REPLY: This information has been added.

Reviewer #2 (Remarks to the Author):

I have reviewed with great interest this manuscript on mite feeding physiology, it is a great piece of work, with a lot of experimental efforts. This study tackles an important yet unresolved question on mite physiology in the honey bee, with a partial comparative approach on varroa and *tropilaelaps*. Using physiology and omics it shows that varroa are most likely to feed primarily on haemolymph during its reproductive phase, while it preferentially eats from fat bodies during dispersal. I have several comments that would prevent publication of the manuscript in its present form.

REPLY: We thank the reviewer for the general support and address the concerns below.

First, it is presented as a comparative approach between *Varroa* and *Tropilaelaps*, but the comparison is not maintained throughout the results so I do not think that the title and claims fully match the work. In fact *Tropilaelaps* is only studied in the staining experiments, targeted on the reproductive phase. So unless I am wrong no data is presented for this mite on the dispersal phase. There is also no data on the proteomics and metabolomics. So I would center the study around varroa mainly, and maybe have a little spotty comparison with *Tropilaelaps*.

REPLY: There is no distinct dispersal phase in *Tropilaelaps* and therefore the study cannot be completely parallel in the two species. However, it is true that we could support our conclusions further with additional data from *Tropilaelaps* and therefore we have performed an additional proteomic analysis to compare honey bee-derived proteins found in *Tropilaelaps* with those found in dispersing and foundress *Varroa* and with honey bee hemolymph. The results support our argument that reproducing *Varroa* and *Tropilaelaps* primarily feed on hemolymph and that the diet of dispersing *Varroa* fundamentally differs.

I also find a real shame that no staining approach has been done on the dispersal phase. In the title you explicitly mention the "life-history stage" approach, but why doing it only for the proteomics and metabolomics?

REPLY: We chose not to directly replicate the experiments by Ramsey et al. (2019) because they provided extensive results on feeding sites during the dispersal phase and we have no reason to doubt these results.

L39: *Varroa destructor* should be in italics

REPLY: Agreed and corrected.

Introduction: it reads very well and is well documented.

L.95: I believe the the dispersal phase can be (much) longer than seven days. I had 4-6 days long as an average, but up to several months in Northern climates.

REPLY: The statement was revised according to the reviewer's suggestion and literature.

L.98: I would state both reproductive rates to highlight the difference more clearly.

REPLY: The reproductive rates of *Varroa* and *Tropilaelaps* vary among populations and depend on which type of brood (worker vs drone) is parasitized and whether single or multiple infections are considered. Spelling all this out would make the statement very clunky and not compatible with NCs recommendation for clear and concise writing.

I would rearrange the first part of the results, because at the moment it is not very clear what the focus is, and why (varroa only, varroa and tropilaelaps). You will need to justify more clearly why the first part is done on both, and the second and third parts only on varroa. At the moment it is confusing.

REPLY: As we have added the proteome analysis of *Tropilaelaps*, we have restructured the results & discussion section to clarify the distinction between *Varroa* and *Tropilaelaps*.

I did not understand what the input is of the first experiment (Feeding sites of Vd on honey bee pupae) for the question. It needs to be better justified, or removed.

REPLY: We tried to indicate the relevance of this experiment with the sentence "The source of nutrition for parasites is largely determined by the site of parasitism on the host, so we first investigated the feeding site of *Varroa* mites on honey bee brood." This argument is now additionally supported by our own investigation of the bees' abdominal fat body, complementing earlier studies that are cited in the text.

I would then make a comparative section on the food ingested by both mites while in capped brood (Fig 3 and Figure 5). Then I would develop Fig 4, which is a nice comparative conclusion.

REPLY: To clarify the comparative aspect, we have restructured the entire section (see comment above). This has forced us to split up former Fig. 4 into two smaller figures, which does not prevent our comparative conclusions but may be a little less convenient for the reader.

To make to story complete, I miss the staining done for mites in the dispersal phase. If for this part you only base your conclusions on the comparison with the literature (Ramsey et al) it should be clearly stated. For me staining of dispersing varroa and tropilaelaps need to be included.

REPLY: We now cite more explicitly the study that characterized feeding sites of *Varroa* on adult bees (Ramsey et al) and of *Tropilaelaps* on brood (Phokasem et al).

L250-252: it is such a shame not to have the equivalent of Fig 2 for tropilaelaps.

REPLY: The results have already been published, see Phokasem et al. 2019.

L 287 and M&M section: the fact that you have chosen mites from purple eyed pupae is a big issue to me, and probably biases the results a lot (it has been shown previously that mite transcriptomics

varies a lot along the bee capped development). I understand the choice to sample juveniles (that do not bring that much to the story - I would place these aspects as supplementary figures), but to sample a reproductive mite I would have targeted anything from prenupa to white eyed pupa. I believe that at purple eyed the foundress is not really reproducing anymore sensu stricto. So you probably miss the key window, purple eyed.

REPLY: We agree with the reviewer that we definitely did not capture the initiation of reproduction, which was also not our intent. By choosing *Varroa* on purple-eyed pupae, we could select only foundresses that were successfully reproducing, which was important to us (as specified in the M&M section). It also allowed us to collect juveniles simultaneously, which are not central to our argument but nicely corroborate the metabolomic distinction between foundress and dispersing adult *Varroa*. Most importantly, *Varroa* are still actively producing and laying eggs during this time of host development (Rosenkranz et al. 2010; Dietemann et al 2013) even though these eggs usually cannot complete development and emerge as adults. Which may be the reason why Mondet et al., (2018) refer to such mites as “post-laying mites” and the reviewer refers to “reproduction sensu stricto”. However, the ultimate fate of offspring does not invalidate the argument that the *Varroa* mothers are actively laying eggs at this point.

L.480-486: can you give insights into the success rate? Survival? Mite fertility? Fecundity? To show that the in vitro system really mimics what happens in the hive.

REPLY: We did not collect these specific data here because this technique is an established practice in our lab with typical survival rates of 90% and 80% reproduction among surviving foundresses. Usually, about 3 progenies can be collected per reproducing mite, similar to Nazzi and Milani (1994) who invented this rearing method.

L.521-524: in your introduction you state that the dispersal phase is at maximum 7 days. So I believe that by leaving mites for 7 days without brood, you may force the system. I would have picked 4-5 days.

REPLY: As described above in response to a comment by the same reviewer, we needed to (and did) correct that statement about the maximum duration of the dispersal phase (because it can

actually be much longer than 7 days).

Reviewer #3 (Remarks to the Author):

This study throws a new and much needed look at the question of which host tissue do highly damaging ectoparasite mites feed on when infesting honey bees, the major managed pollinator. The authors follow up on a work that led to a paradigm shift, but which remained controversial and not yet accepted widely in the field. This new study aims at using more direct methods (proteomics and metabolomics) to determine the diet of the mite *Varroa destructor*. It also replicated the methods used in the previous work, but on another life history stage of mites and thus provides a more complete picture of the nutritional adaptation of the parasite to the host. The authors conclude that the mites shift diet from using principally fat bodies when infesting adult hosts to hemolymph when infesting honey bee brood during their reproduction phase. They also performed some of the experiment on another ectoparasitic mite, which represent a threat to beekeeping worldwide and discuss the differences in diet adaptation of these two mite species.

A main issue I have with the way the manuscript is written is that the introduction does not set the stage of the study properly so that one wonders why some experiments were done and how they support the main aim. Clear hypotheses and expectations for each experiment are missing. Also, most of the data does not clearly support the conclusions, at least not the way it is presented, because of some apparent circular reasoning and poor connection between the results of the various experiments. An important piece of information missing is the structure of fat bodies in the pupae used for the biostaining experiment. This knowledge is fundamental to assess whether the difference in diet proposed is likely determined (as the title suggests) by host life history stage. Overall, the manuscript does not convince the reader and this could be due to the poor support of the conclusion by the data or by a suboptimal presentation of the data.

REPLY: The perceived disconnect is unfortunate and we have revised the manuscript throughout to improve the clarity for the reader. Furthermore, we have now added the requested study on the structure of the fat bodies during development and show (in further support of our main result) that they only become available during the very last stages of pupal development at the predominant *Varroa* feeding sites.

Major comments

Some of the results presented go beyond the rationale of the study presented in the introduction. The analysis of feces comes out of the blue in the results and it is not clear how studying the omics of the immature mites contributes to the main aim: the determination of diet when feeding on brood.

REPLY: The analysis of feces is important because the excretion of fat body material could be an alternative explanation for our result that mites are predominantly stained green. We stated as much previously but have now moved the interpretation closer to the result presentation to clarify the connection.

The metabolomics comparison of immature *Varroa* provides valuable evidence about their similarity to foundresses and not dispersing adults, which we highlight in the corresponding results section. While this is not essential to our argument, the overall similarity (Fig. 6B) supports the argument from a different angle. To emphasize the usefulness, we have now added that information

also into the figure caption.

Together with the lack of hypothesis statement and expectations making it difficult to determine the strength of the result interpretation, this gives a confuse impression to the reader who does not end the manuscript convinced of the conclusions proposed. The logic of several arguments is not clear, with some appearing circular. This should be improved to better make use of the results presented and increase the impact of the paper.

REPLY: It is not clear to us what the reviewer is specifically referring to and we would like to ask for clarification if this comment is addressing concerns other than the ones specified (and responded to) elsewhere in this document.

The data presented for *Tropilaelaps* derives from the same methodology as used in the previous work on *V. destructor* which was judged as insufficient (L117-119) and prompted the new study. *Tropilaelaps* were not subjected to proteomics and metabolomics to further support the argument of a haemolymph diet. The evidence for *Tropilaelaps* thus appears weak and does not bring the question of the diet of these ectoparasites substantially further. Furthermore, the feeding sites of *Tropilaelaps* were not determined. The *Tropilaelaps* part of the work could be the topic of a separate paper without weakening the study much.

REPLY: We thank the reviewer for the suggestion that the *Tropilaelaps* study is worthy of a separate publication. However, we would prefer to include the comparative aspect in this work and have added a parallel proteomic study to address this concern and strengthen the synergism between the two study systems. We have also reworded the sentence previously in L.117-119 because we do not mean to doubt the actual results presented in the Ramsey et al. study.

L138-140: I do not agree with this conclusion because the current hypothesis that *Tropilaelaps* feeds on brood only is because of anatomical limitations of their mouth pieces. Your argument requires hypothesising a reduction in mouth piece strength following the reduction to a pure hemolymph diet on the brood.

REPLY: We have omitted this sentence because it is not important to the present study and indeed just speculation in which evolutionary order these behavioral and morphological changes occurred.

L119-124, 342-346: whether parasite diet is constrained by external anatomy of the host or a choice of the mite is indeed a very relevant and has not been sufficiently addressed. However, there are not enough details as to when the tissue structure of adults (fat bodies lining the cuticula) is acquired during pupal development. Here you only mentioned a diffuse pattern in metamorphosing individuals and not in the pupal stages tested. This information is fundamental to determine the relevance of the pupal stage you selected for your dye experiments (which by the way is not mentioned in section 4.4) and for the interpretation of your results. If fat bodies are already lining up the cuticula when reproductive mites are feeding on the hemolymph of these pupae, as proposed, then mites can choose their diet and are not constrained by host internal or external anatomy. This is especially important because you show that the feeding site on pupae is virtually the same as on adults.

REPLY: In response to this comment, we performed an additional study of the fat body lining the cuticula at the predominant feeding sites of *Varroa* (i.e., abdominal sternites). As now reported, we

document a gradual build-up with significant fat body only available shortly before the emergence of the adult bee. We report this in the main text but have submitted the actual pictures as a supplemental figure because the characterization of the fat body is an auxiliary confirmation of the previous work of Snodgrass (1956).

On this note, you also should better discuss the incompatibility of your results with Ramsey's et al feeding experiments, which showed that mites need access to fat bodies to better survive and reproduce, while you show they do not have such access.

REPLY: We do not take pride in explicitly criticizing the work of others in public, but the tissue-feeding bioassays of Ramsey et al are marred by very low survival and reproductive rates and our study indeed invalidates some of the previous claims. We have now added a sentence contrasting our study to Ramsey in the conclusions on account of this suggestion.

L134: I argue that your experimental design does not allow you to suggest that hemolymph diet is more beneficial to mite offspring. You have not manipulated their diet to show this.

REPLY: We agree that we do not show that it is superior and have omitted this statement.

L278-283: these proteomics results are not discussed and because of the lack of hypothesis or expectations presented for these analyses in the introduction, I have no idea how they contribute to the main aim.

REPLY: We have now added an explicit prediction to the start of the combined results/discussion section, explicating that we predict honey bee proteins in *Varroa* to reflect the food source and therefore would be distinct between dispersing and foundress *Varroa*, and be biased towards hemolymph proteins in foundresses.

L331-333: what allows you to determine that the differences observed with this proteome comparison supports the hemolymph feeding by foundresses? To me this sounds like an assumption and circular reasoning, not a piece of evidence. Protein metabolism is determined by the physiological needs (here oogenesis probably), not by the type of diet as long as proteins are present. These proteins could very well come from the fat bodies.

REPLY: It seems that our presentation here was not clear enough and the reviewer is mixing two separate arguments. We have thus revised this paragraph (formerly Lines 312-333) to clarify that it does not try to distinguish honey bee hemolymph versus fat body as food source (by comparing honey bee proteins found in *Varroa*; which is presented and discussed in the previous three paragraphs), but instead investigates the consequences of the different diets for the physiology of *Varroa* (by presenting and discussing which *Varroa*-endogenous proteins are differentially present in dispersing or foundress *Varroa*).

L334-336: same comment, you assume hemolymph feeding, the proteomics data does not show it!

REPLY: We disagree. The proteome analysis of host-derived (honey bee) proteins provides the following evidence for hemolymph-feeding of foundresses compared to dispersing *Varroa*: 1) Reproductive foundresses and dispersing *Varroa* contain distinct sets of (undigested, otherwise they would not be identifiable) honey bee proteins. 2) The set of honey bee proteins in foundresses contains 78 (=74%) honey bee hemolymph proteins. We don't know how we could provide more

direct evidence.

L364: the strength of the metabolomics approach in identifying the diet type of each mite life history stage is weakened by the mere ‘usual reflection’ of diet difference in the metabolome. Excluding that the metabolome cannot be different on the same food source would be more powerful to support your conclusion. However, Fig 7C showing different patterns for foundress mites and its offspring, show a different metabolome on the same diet is actually possible.

REPLY: We agree with the reviewer that there are metabolomic differences between foundresses and offspring and this would be expected based on the very different biological processes that happen during adult reproduction and juvenile development. Thus, the argument here is about relative overall similarity among metabolomic profiles, which is reflected in the clustering of Fig. 6B (and described and discussed in the text). Since metabolites are not species-specific, an analysis that parallels the preceding proteomic analysis is unfortunately not possible.

Other comments

Title: The title could be improved to give a more representative idea of the work. Replacing ‘dietary alternation’ by ‘diet’ would improve readability. However, because you have not shown that the mites do not have the ability to choose what they feed from, you cannot write that the stage determines the diet (this is only a hypothesis at this stage, see your L171-172). You can only state that the diet differs between the stages. See major comment on the structure of fat bodies in pupae and on anatomic constraints to diet.

REPLY: Following the reviewer’s suggestion, we have made the title more descriptive. To describe that *Varroa* mites alternate between different food sources while going (repeatedly) through their life cycle, we consider the word “alternation” essential because simple “change” does not capture the biological reality. To avoid the title to be overassertive, we have purposefully used the word “determines” instead of “causes”. This does not imply an active choice. Furthermore, we have now added the requested characterization of the structure of the fat bodies in pupae.

I suggest referring to the species by their full scientific names throughout the manuscript. There are several *Varroa* and *Tropilaelaps* species in these genera, but only one of each has been experimented on here. When several species are meant, the genus name followed by ‘spp.’ is required.

REPLY: We have followed this suggestion and inserted full species names throughout the manuscript where applicable.

Abstract:

L16: should probably be ‘genera’ instead of ‘genus’

REPLY: Thank you, corrected.

L25: the manuscript does not present proteomic data for *Tropilaelaps*, this sentence is thus misleading.

REPLY: We have now added proteomic data for *T. mercedesae*.

L32-33: your experimental design does not allow you to show the adaptation itself, a more careful formulation is required.

REPLY: We agree and have replaced “that is adapted to” with “in accordance with”.

L38: I am not convinced that the issue is resolved, especially for *Tropilaelaps*.

REPLY: This sentence specifies *V. destructor*. We are not aware that there is or was a controversy in *Tropilaelaps* spp.

L39: *V. destructor* should be in italics

REPLY: Corrected.

L33, 40, 43, 365: can one speak of specialization or of evolved a strategy if the diet choice is constrained by the host internal anatomy? This possibility is not excluded by your work. See major comment on anatomical constraints.

REPLY: We certainly think so. In our opinion, a deliberate choice is not necessary for evolution (of specialization or otherwise) to occur and host constraints can lead to evolution (e.g., Harvey 2005: <https://doi.org/10.1111/j.1570-7458.2005.00348.x> or Noort et al. 1996: <https://doi.org/10.1111/j.1365-2699.1996.tb00003.x>).

Introduction:

L61: ‘Eastern’ instead of ‘Easter’

REPLY: Corrected.

L78: the female is not gravid (ie bearing oocytes) yet when she enters the cell, see also your text L91.

REPLY: We agree that “gravid” was a poor word choice and have replaced it what we really mean: “mature”.

L76-88: contain details not required for the understanding of the rationale of this study and should be omitted. Eg initial egg developing into a male etc.

REPLY: We prefer to keep the basic description of the mites’ life cycles in the introduction for the non-specialist reader and consider the comparative aspects (e.g., *T. mercedesae* use multiple feeding sites”) particularly valuable.

L83: ‘rapidly’ is too vague, please be precise, but see previous comment.

REPLY: Without going into much more detail (i.e., female vs. male offspring, *T. mercedesae* vs. *V. destructor*, and even variation among studies; Martin 1994 & Anderson et al. 2013), it is not possible to be more precise.

L86: only male and female offspring will mate

REPLY: In response, we made this phrase more specific.

L104: it is unclear to me how maturation links to reproduction mentioned earlier

REPLY: This is intended as a contrast between the reproductive stage and development as distinct life cycle stages with distinct physiological profiles that match the biological function of these stages (to foreshadow our own work). We have now replaced the “while” conjunction with “whereas” and

“maturation” with “development” to clarify.

L101-110: at this point it is difficult to determine why this information is given. It is useful for the study? Will this information be used again? If yes, this should be made clearer. Clarifying the rationale, hypotheses tested and expectations of the study will help making sense of this information. Alternatively, please consider giving this information in the discussion as you put your results in perspective.

REPLY: See previous comment. Furthermore, we have additionally rephrased this paragraph to make it more relevant background information to interpret our own study.

L117-119: Please specify what information is lacking to show more directly which diet is used by the mites and explicitly mention how the study fills this gap.

REPLY: We have rephrased this sentence to avoid directly criticizing Ramsey et al. and connecting it better to our study.

L125 and following: I suggest removing the results from this section.

REPLY: We agree and have moved the entire paragraph (with modifications to state our hypothesis and derived predictions more explicitly) to the introductory part of the “Results” section.

L135: please clarify which stage is meant here

REPLY: We clarify “life stage” and then explain which life stages do what in the subsequent sentence.

L150: at this stage, we do not know if the nutrition is constrained by the feeding site, see your L171-172. This kind of information relating to the rationale of the study should be presented in the introduction, see major comments.

REPLY: We have changed the sentence structure to put emphasis on what we did, but think that a little bit of recap justification here is important to enhance the reading flow.

L154: a confirmation requires a citation for the work confirmed

REPLY: We agree and have added the citation.

L157-159: can this not be excluded with results of citations 42 and 43?

REPLY: Both of these studies show a rare occurrence of multiple wounds and we just like to highlight the limitation of our study (by not studying multiple age groups of pupae).

L160: please provide figures with the same level of precision throughout (one decimal) for consistency.

REPLY: Thank you, corrected.

L164-166: why would this be useful? Please expand the discussion and do not let the reader guess the importance of this information.

REPLY: We have added a sentence to explain the potential benefit.

L173: ‘...mites feeding on brood’. Varroa mites do not feed on uncapped brood, so no need to

mention capping here.

REPLY: The title has been changed to be more precise.

L174-182: contains superfluous repetitions of the introduction and hypotheses which should already be given in the introduction.

REPLY: We agree that some (but not all) of this information can be cut and have accordingly shortened this section.

L178: I disagree, they feed on the same host, but at a physiologically different stage.

REPLY: We specified our intended meaning better to clarify that we don't mean different species.

L186: not only larvae, pupae too.

REPLY: Yes, we agree and therefore changed the word to "brood".

L187: how does this validate biostain quantification? The logical link is unclear to me.

REPLY: We thank the reviewer for pointing out the imprecise statement and have changed it to specify our intended meaning.

L190: pupae do not feed. I suggest precisizing which tissues are stained by which dye so that the reader does not need to consult the text to find out.

REPLY: We changed the statement to "Photos show honey bee pupae that fed as larvae on biostains" and explain which color stains what tissue, as suggested.

L197: the consistency of the staining is not shown in Fig 3 (only one representative individual chosen) and repeats L184-185.

REPLY: Yes, the representative figure is intended to illustrate our observations described in the text and therefore needs to repeat the message. Quantitative analyses are provided below and illustrated in the next figure.

Fig.4: the readability of this figure would be increased by separating the two mite species. Here too reminding, in the caption, which dye labels which tissue would help the reader. Please also mention the pupal stage on which the mites fed.

REPLY: Following this suggestion and the restructuring of the manuscript, we have now split figure 4 into two (according to mite species).

L228-233: the visual assessment of fluorescence intensity is not reliable and the quantitative comparison should be used here. The latter does not show a weaker intensity of foundresses compared to deutonymphs. Although no statistics are available to quantify this difference or lack thereof.

REPLY: The *in-situ* staining is a separate experiment from the quantitative measurement of fluorescence and both contribute to our results. The description of the *in-situ* staining is qualitative and we took care to word the conclusions cautiously. However, in response to the reviewer's comment, we have added statistical tests to the quantitative comparisons (*V.d.* and *T.m.*).

L238: I am not convinced by this argument because, as you mention L208, feces are a mixture originating from all stages.

REPLY: For the ratio of deutonymphs (666) to increase to that of adults (1690) by preferential excretion of lipophilic Nile red requires some very improbably conjectures, given that the feces overall have a ratio of 161. IF there was total selectivity and adults would only excrete Nile red, every mg of adult feces would have to be outweighed by >160 mg of juvenile feces AND we would have to also assume that juveniles display complete selectivity in the opposite direction than adults (only excretion of Uranine), which is next to impossible in this mental construct because deutonymphs have also a higher ratio than protonymphs.

L242: is an interpretation of the results and rather belongs to the main text

REPLY: Similar to the Fig. 3 caption, we intend to title this caption with the main message of the figure for the casual reader.

L248-253: belong to the introduction and should support the rationale for the study.

REPLY: We consider this part of the discussion to put our results in context and facilitate their interpretation.

L254: indicates

REPLY: Thank you, corrected.

L258-259: the results just mentioned refer to fig 4 which shows the results of the quantitative analyses, and not the visual results mentioned here.

REPLY: The reference to the figure was omitted.

L261-262: this is hard to see as the figure shows the values of these parameters and not their ratio. Could the figure only show ratios? Or add another panel with the ratios?

REPLY: Since we list the ratios in the text, we do not want to repeat this information in the figure. The figure is now also split, which makes direct visual comparisons difficult. To address the reviewer's concern, we omit the reference to the figure here and focus the reader on the text information.

L263-271: interesting but off topic regarding the main aim of the study in my opinion.

REPLY: We agree with the reviewer that this finding is not central to the main aim, but we consider of sufficient interest to the general topic that we prefer to keep it, unless the editor requires shortening of the manuscript.

L273-278: again this should be part of the introduction to support the rationale of the study. In addition, It is not clear how you identify proteins in Varroa that can readily be identified before complete digestion. Is this ahead of the analyses or post hoc? I do not understand the logic of how the comparison between dispersing and reproductive mites can guard against methodological biases. Which biases by the way?

REPLY: The placement of the specific rationale for this particular analysis facilitates the

interpretation of the results and their discussion which directly follow. We have added language to the methods description to clarify how undigested honey bee proteins can be identified in *Varroa* by our proteomic methods by matching the data against a combined database of *Varroa* and *Apis* proteins. We have also added a qualifier to explain that the potential bias we are guarding against would be a preferential representation of hemolymph- or fat body-derived proteins (e.g., one food type could be digested more easily than the other and therefore be less identifiable). This is probably not the case, but we cannot 100% exclude it *a priori* and therefore safeguard against it by comparing both types of adults.

L285-288: this inconsistency deserves an explanation!

REPLY: There is no inconsistency, but we have replaced “Vitellogenin” with “Honey bee vitellogenin” to preempt misunderstandings.

L295: the use of ‘respectively’ in this sentence should be revised. Respectively to what?

REPLY: The sentence has been rephrased to clarify.

L298-299: what should be concluded from this difference? Please discuss.

REPLY: We have added our conclusion from this difference in an additional sentence.

L303-306: could this difference not be due to a higher consumption of the founding mites to sustain oogenesis? Your hypothesis implies that dispersing mites store MRJPs acquired during the previous reproductive cycle and do not metabolise it. Is this possible?

REPLY: MRJPs are found in the fat body of adult bees and in the hemolymph of brood. Concordant with our hypothesis, we find these honey bee MRJPs in foundresses and dispersing *Varroa* because foundresses feed on brood hemolymph and dispersing *Varroa* feed on fat body of adult bees. We therefore have not implied the storage of MRJPs.

Fig 6 would be more reader friendly with the mite type spelled out instead of abbreviated. Most of the text is of too small size for easy reading. Please mention the host developmental stage from which foundresses were collected in the caption.

REPLY: We have now spelled out the mite type. The ultimate size of the figure during production will have to reflect the level of detail that is presented in this figure and we will ensure readability to the best of our control with the publisher. We have added a reminder of developmental host stage in the caption.

Fig 6C: this is expression data, gene expression? Not protein quantity? Is this the correct terminology here and in the text? Against what was the data normalized?

REPLY: It is protein data and therefore we have changed the wording to “abundance”. The normalization procedure is provided by an inherent algorithm of the MaxQuant (v.1.6.17.0) software. We have thus added this information to the method description.

L309-311: sorry, I do not follow your logic here. Something may be missing to make sense of this.

REPLY: We have rephrased the sentence to clarify the logic better.

L340: the reader expects a more detailed explanation for this inconsistency! A new paragraph should be started at the end of this sentence. And a better transition is needed to link the resulting paragraphs.

REPLY: While we prefer not to split the paragraph because it deals all with the quality and quantity of available hemolymph versus nutritional needs and other factors that may affect food selection, we follow the reviewer's advice to add a transition phrase here. As for the inconsistency with Ramsey et al. we want to note here the very low survivorship and reproduction of the mites in this experiment even under the best conditions, but we want to refrain from publicly criticizing that published work (but see next comment).

L346-351: this appears to be off-topic. I do not see any logical link with the previous sentences and the aim of the study.

REPLY: The nutrient balance has a lot to do with the external environment and we therefore included this relevant discussion. It might explain the "inconsistency" mentioned in the previous comment and we have now added a sentence to point this out.

L363-367: again these are details I would expect in the introduction associated with an hypothesis tested and expectations to support the use of either diet type.

REPLY: We shortened and reformulated this section to include the hypothesis and prediction.

Fig 7: Please mention host stage on which the foundress and offspring were collected. Most text is too small. A) The meaning of QC is not given.

REPLY: We apologize for the small font in its current format and hope that the production team will do the details justice. Host stages are now specified in the legend, as is the explanation for QC (=quality control as described in the methods).

L415-427: this appears off-topic and does not contribute to the main aim of the study in an obvious manner.

REPLY: We think the corresponding results deserve discussion, even if they are not directly "useful" for our central hypothesis.

L425: emerges from

REPLY: Thank you, we corrected the error.

L439: a few words explaining why this is so would be desirable, instead of just citing a study.

REPLY: We have followed this suggestion and added clarification.

L439-442: What do you mean with 'selective environment'? I do not follow the logic of this argument. What does this situation have to do with the source of their diet?

REPLY: We have reworded this sentence to clarify the argument.

Methods

L485: please mention at which stage exactly. In addition, knowing the structure of fat bodies at this stage will help supporting your claims regarding their relation to mite diet, see main comments.

REPLY: We have now added this information as requested. In addition, we have investigated the

structure of the fat body at the feeding site and provide the additional results.

L517: omit 'directly'

REPLY: Changed according to the suggestion.

Reviewer #4 (Remarks to the Author):

I appreciated that the Authors analyzed Varroa feces during their biostaining experiment/analysis. Compared to the study by Ramsey, et al. (2019), which did not present analysis of feces, the current study considers the rapid food processing of Varroa. Later (line 270 -271) the Authors cite research suggesting rapid food processing in *Tropilaelaps*. Are there similar data available for Varroa?

REPLY: As far as we know, there is no suitable study to cite.

The Authors had the foresight to preempt Reviewers' critiques of the interpretation of their results. In several points (lines 158, 166, 171, for example), the authors place caveats on their results. This may have been intentional, but if not, doing so should help speed up the review process and makes it more difficult for Reviewers to offer criticisms (this is a good thing!).

REPLY: Thank you, we tried to be as cautious in our interpretation as possible.

The results from the metabolomic profiling analysis (as presented in Figure 7) were very convincing for the differences in nutrient sources for reproductive and dispersal stage Varroa mites.

REPLY: Thank you, we agree.

Lines 19 – 20: The Authors should specify Varroa here...since their series of experiments, as reported in the manuscript, were focused on Varroa, and not *Tropilaelaps* mites. For example, there was no mention of proteomic analyses done for dispersal stage *Tropilaelaps* mites.

REPLY: We agree and have rephrased the abstract, in addition to providing additional proteomics results on *Tropilaelaps*.

Lines 89 – 92: It seems that a citation(s) is needed here. Was all the information for *Tropilaelaps* mites reported here directly researched by those cited in (27)?

REPLY: We have added relevant citations.

Lines 129, 130: Here the Authors mention mites are feeding on larvae. Although this may be true, later in the manuscript (e.g., line 137) they switch to using 'brood', and conduct the research reported in the study using specifically, honey bee pupae. Later, on line 186, 'larvae' is used, but in the figure just below this line, images of pupae are shown. The switch between using this-or-that word may be intentional, but it reads awkwardly. A related point to this, lines 480... gives methods for rearing hosts for Varroa foundresses, but I was confused at what stage (instar) the larvae are placed in the capsules. Can the Authors provide more details here?

REPLY: Only larvae can be stained because they consume food. But the stains persist during the non-feeding pupation stage, which serves as host to *Varroa* spp. In contrast, *Tropilaelaps* have also been observed feeding on bee larvae. We have now revised the manuscript to use the terms "brood",

“larvae”, and “pupae” more carefully.

Lines 152 – 154: Results from vital staining wound analysis are given here for Varroa. Given that the Authors had presented introductory information for Tropilaelaps feeding site preferences (line 81), it would be nice to see here a concise comparison between results for Varroa and those previously reported for Tropilaelaps. Also, how does Varroa feeding site preference compare between honey bee pupae and adult hosts? What can be said about this?

REPLY: We have added a comparative sentence.

Line 201: ‘labeled bees’...do you mean ‘brood’ or ‘larvae’?

REPLY: Thank you, we have now specified “pupae”.

Line 214 (Figure 4): Was it possible to present fluorescence intensity of samples on a weight basis? If possible, how would doing so change the results presented in this figure?

REPLY: Unfortunately, we do not have the exact weight measurements of the samples we took. Average estimates could be used but this would be inaccurate. Furthermore, our main purpose to compare the Uranine and Nile red staining in each sample relative to each other would not be affected by adjusting the fluorescence signal to weight.

Line 248: It would have been great to have tested this hypothesis by conducting proteomic/metabolomic analysis for dispersal stage Tropilaelaps...maybe the Authors are saving this for a subsequent paper?

REPLY: We have now conducted an additional proteomics analysis of *Tropilaelaps* and report the results here. It is from the reproductive stage because *Tropilaelaps* does not have a notable dispersal stage.

Lines 257: From Figure 4, I am not sure this statement is accurate. The different stages of Tropilaelaps look as different as those of Varroa.

REPLY: We have now specified that the ratio is less variable in *T.m.* (ranging from 55-205) than that of *V.d.* (ranging from 309-1690).

Lines 312 – 315: Were the numbers reported here for those significant after correction for multiple comparisons? Same question for lines 355 – 357 (Figure 6 caption).

REPLY: Yes, we report number of proteins with significant differences after the Benjamini-Hochberg procedure at 5% FDR, which corrects for multiple comparisons. We have now added the number of significantly different proteins in the caption of Figure 6 (now Figure 5) to improve clarity.

Line 521: What stage were the brood on these open brood frames? Please give this information. Could Varroa have fed on these larvae during the seven days?

REPLY: Only capped brood frames were removed (to avoid foundresses and newly developed mites from mixing into the population of dispersing mites that we wanted to sample. Open brood of all ages was present and theoretically could have been parasitized by *Varroa*. However, this has never been observed in hundreds of *Varroa* studies, leading to the universally accepted view that *Varroa*

do not feed on brood that has not yet been capped.

Line 784: Check this citation. I think it should be S.D, Ramsey?

REPLY: We corrected it to S. D. Ramsey et al.

I believe the captions for Figures S1 and S2 should be more elaborate to include information for which biostain was used. They should read more like the captions given for Figures 3 and 5.

REPLY: We have added more complete figure captions to all supplemental figures.

S1: *Varroa destructor* ingests large amounts of honey bee pupae hemolymph during their reproductive stage. Photos show dorsal views of the specimen in Figure 3 of the main text. Protonymphs (A–D), deutonymphs (E–H) and adults (I–L) of *V. destructor* are biostained after feeding on dually biostained honey bee pupae. Far left: brightfield, center left: fluorescence from Nile red (lipophilic, labeling predominantly fat body), center right: fluorescence from Uranine (hydrophilic, labeling predominantly hemolymph), far right: superposition of all three channels. In addition, corresponding photos of feces (M–P) are presented. Within each photo, the specimen on the left is an unlabeled control and the one on the right depicts a representative labeled sample. All scale bars represent 1 mm.

S2 (formerly S3): Protein-protein interaction enrichment analysis using the differing *Varroa* proteins identified eight clusters associated with four functions. One large cluster is associated with “Ribosome”, while 2-3 smaller clusters are associated with the other functions. Most clusters were dominated by proteins that are up-regulated in foundresses (“Fou up” in yellow) and few of these proteins were more abundant in dispersing *Varroa* (“Dis up” in blue).

S3 (new): The distribution of fat bodies in the abdomen of developing bees. Photos show biostained honey bee pupae or adults in brightfield (first column), the dissected abdominal sternite in brightfield (second), fluorescence from these abdominal sternite samples associated with Nile red (third) and Uranine (fourth), and all three sternite images merged together (fifth). All scale bars represent 1 mm. Photos of white-eyed pupae (A1–A5), pink-eyed pupae (B1–B5), purple-eyed pupae (C1–C5), grey-wing pad pupae (D1–D5), newly emerged adult worker (E1–E5), and newly emerged adult control (Ctl1-Ctl5).

S4: Metabolome comparison among protonymphs (Pro), deutonymphs (Deu), foundresses (Fou) of *Varroa destructor*. (A–C) OPLS-DA analysis, expression analysis, and KEGG pathway enrichment in the comparison between Pro and Fou. (E–F) OPLS-DA analysis, expression analysis, and KEGG pathway enrichment in the comparison between Deu and Fou. (G–I) OPLS-DA analysis, expression analysis, and KEGG pathway enrichment in the comparison between Pro and Deu.

S5 (formerly S2): *Tropilaelaps mercedesae* ingests large amounts of honey bee pupae hemolymph during their reproductive stage. Photos show dorsal views of the specimen in Figure 7 of the main text. Protonymphs (A–D), deutonymphs (E–H) and adults (I–L) of *T. mercedesae* are biostained after feeding on dually biostained honey bee pupae. Far left: brightfield, center left: fluorescence from Nile red (lipophilic, labeling predominantly fat body), center right: fluorescence from Uranine (hydrophilic, labeling predominantly hemolymph), far right: superposition of all three channels. Within each photo, the specimen on the left is an unlabeled control and the one on the right depicts a representative labeled sample. All scale bars represent 1 mm.

S6: Proteomic analysis of honey bee (*Apis mellifera*) – derived protein found in *Tropilaelaps mercedesae*. Overlap with the equivalent analysis in *Varroa destructor* is showing that most are shared and the overlap is significantly higher with proteins that are uniquely found in *Varroa* foundresses (56 of 105) than with dispersing *Varroa* (0 of 15).

Figure S2: Why were representative images of *Tropilaelaps* feces not shown?

REPLY: Unlike the *Varroa* mites, which have a fixed defecation site in the capped cells, the small *Tropilaelaps* feces are scattered in the cells (or in our case, capsules) and cling to the wall, making it difficult to collect a “photographable” mass.

REVIEWER COMMENTS

Reviewer #2 (Remarks to the Author):

Thank for your revised version of the manuscript. I found all comments to have been addressed, even though some responses were not integrated in the new version of the text, and are only present in the "response to reviewers" document.

Reviewer #3 (Remarks to the Author):

The authors have improved the text, which now presents the rationale much more clearly. Hypotheses are stated and the reader now understands what was done for which reason. Well done! The authors also added new data on *Tropilaelaps* proteomics and host internal morphology to support their arguments, this is remarkable.

Most my comments were addressed in a satisfactory manner. I nevertheless still have an issue with the main proteomics result as some choices made for comparing ingested proteins and reference hemolymph proteome do not appear justified to me. I also have some minor comments, which I hope can further improve the manuscript.

In absence of line numbering, I copy the manuscript text below and append my comments with a '>' sign under the respective text.

Major comments:

Overlap with a representative honey bee hemolymph proteome (50), was significantly higher ($\chi^2 = 10.3$, $p = 0.001$) for the proteins that were specific for foundresses (78/105 = 74.2%) than for proteins that were specific for dispersing mites (5/15 = 33.3%).

> This is a central argument, the whole proteomics approach and conclusion is based on this and thus needs to given more prominence and clarity. I suggest making a figure to illustrate this result, which could complete Fig. 5a or replace Fig. 4. The latter is not very informative (because the more telling ratios are given in the text) and could be in the SM. The reader would also like to see a discussion on the relevance of this result for the research aim. Also a discussion about why is it not 100% and if a non 100% result is weakening your interpretation would be needed.

> Because this element was diluted among other results without being discussed, I previously failed to recognize its importance regarding the hypothesis tested. I now wonder why were only the proteins specific to either the dispersing mite or the foundresses considered for the match with hemolymph proteins and not the 167 proteins common to them, knowing that these are host proteins and not varroa proteins? Also host protein common to the two mite types should be considered to determine the food source, unless I missed something, which should be mentioned in the manuscript for other readers not to wonder as I did. Including these 167 proteins may shift the % match with hemolymph proteins between the mite types. In the extreme case of the 167 common proteins all belonging to the hemolymph, the % match become 90 for foundresses and 94 for dispersing mites!

> If I understand correctly, all the protein identified (272 for reproductive and 182 for dispersing individuals) are honeybee proteins (i.e. whole body proteome: hemolymph plus other tissues) and in the next step, only the hemolymph proteins are looked at. Why not restricting the comparison to the relevant hemolymph proteins directly? This would be less confusing. Would any important information be lost omitting whole body host proteome? From your argumentation in the text, I do not recognize any.

Among the 78 proteins that were uniquely found in *Tropilaelaps*, 55 proteins (70.5%) overlapped with a representative honey bee hemolymph proteome (50).

> Why are only these *Tropilaelaps* specific proteins considered for a match with host hemolymph

proteins? Those host protein also found in *V. destructor* if it feeds from the same source also allow an interpretation regarding protein origin for *Tropilaelaps*. Considering the proteins also found in *V. destructor* might change the % match.

> And same comment as above: why not focus straight away on hemolymph proteins?

However, the quantitative difference (in hexamerins) to dispersing *V. destructor* demonstrates the utility of our proteome approach in general because the mite proteomes reflect the much higher expression of hexamerins in honey bee brood than in adults (60, 61).

> I see no obvious quantitative difference, if any at all, between dispersing and foundresses in Fig 5c. Are there statistics to support this claim?

Fig 5(C) Major royal jelly proteins and hexamerins from honey bees were consistently more abundant in foundresses than in dispersing mites.

> I see no consistent pattern here, see previous comment. Some MRJPs appear equally abundant in both groups. Are there statistics to support this claim?

minor comments:

Title: It is a pity to restrict the work to *V. destructor* now that you completed the study with the previously missing *Tropilaelaps* proteomics. what about being more inclusive? Just a suggestion: Life-history stage determines the diet of ectoparasitic mites on their honey bee hosts

Abstract:

Biostaining and proteomic analyses were corroborated by corresponding experiments on *Tropilaelaps mercedesae*, a mite that only feeds on brood and has a strongly reduced dispersal stage.

> The ultimate aim is not to corroborate analyses, the aim of this comparison should be rephrased as stated in the introduction.

Introduction

> What are direct experiments? This should be rephrased more accurately.

Results and discussion

> Aims and hypothesis stating as well the mention of which methods used to reach the aims classically belongs to the introduction. The reader would also appreciate a mention of the secondary aims, which justify some results section on proteomics (physiological state of dispersing and foundress *V. destructor* with four figures) and metabolomics (speculative view into the developmental physiology), which are not directly linked to the main aim.

The results on feeding sites, however, do not fit in this section. Since the feeding site on pupae is known at least since Donze and Guerin 1994, your results confirming or precisising this are not required to present the rationale of the next steps of your study.

the majority of the life cycle of *V. destructor* and *Tropilaelaps* spp. consists of the reproductive stage during which these bee mites feed on pupating honey bees

> This statement is not totally correct because of long winters spent in the dispersal phase for *V. destructor*.

Our test used the same methodology of labeling host tissues with two fluorescent biostains, the hydrophilic Uranine and lipophilic Nile red, to distinguish the primarily hydrophilic hemolymph and lipophilic fat body in the mites after feeding (33).

> Same methodology as what?

Similar proteomic profiles of ingested food and specific host tissues provide molecular evidence of feeding preferences. We performed this analysis comparatively on reproductive and dispersing *V. destructor* to guard against potential biases that might favor the identification of proteins that are either derived from the honey bee fat body or hemolymph.

> the explanation for this precaution is better phrased in your rebuttal letter:

the potential bias we are guarding against would be a preferential representation of hemolymph- or fat body-derived proteins (e.g., one food type could be digested more easily than the other and therefore be less identifiable).

> it is however not clearer to me how the comparison between foundresses and dispersing individuals guards against this bias. Digestion ability could vary between life stages.

> Fig 5(B) legend, 'non-significant' rather than 'insignificant'.

(D) Differentially expressed proteins of *V. destructor*'s own proteome (non-honey bee derived) clustered perfectly between foundresses and dispersing mites.

> What is a perfect clustering? This should be rephrased.

> Fig. S3: how come the fourth column shows no green staining? These individuals have no hemolymph? The explanation becomes clear later in the methods:

The abdominal sternite was cut out and briefly washed with PBS. Then, the inner surface of the abdominal sternite was photographed

> In this case, no need to show or mention the Uranile column, what it is supposed to stain is gone!

The results indicated a high stability and reproducibility of our results because quality controls and other sample groups were tightly clustered together and separated from each other (Fig. 6A).

> 'tightly clustered together and separated from each other' sounds contradictory and would benefit from reformulation.

> Fig 6(C) legend, 'non-significant' rather than 'insignificant'.

although protonymphs, deutonymphs, and foundresses of *T. mercedesae* appeared not as different from each other as experimental groups of *V. destructor*.

> This observation deserves a discussion, even if speculative: why is it so?

However, these ratios in *T. mercedesae* varied less and exhibited a pattern that was inverse of what was observed in *V. destructor* with the highest ratio in protonymphs and the lowest in foundresses.

> Same comment as above. Below you only discuss the large differences in absolute values between the species.

Conclusion:

Our findings are incompatible with the claim that *V. destructor* rely on honey bee fat body (33)

> This statement should be nuanced by adding 'primarily' and/or by specifying 'during all life stages'.

The dispersal, reproductive, and developmental stages have distinct nutritional needs

> This may be true, but your work indicates that it is not the need that determines the source of the food, but its accessibility. See your explanation on why the less nutritive hemolymph suffices to sustain reproduction. See also next comment. Is there a clinch or compromise here or not? This is unclear to me.

These host stages match *V. destructor* requirements for mobility with differences in food availability

because the brood is immobile and filled with proteinaceous hemolymph, while the adults are highly mobile and contain more localized fat body tissues.

> Please better explicit the connection between mobility and food type. Proteinaceous hemolymph sustain reproduction metabolism and fat body tissues energy metabolism? Is this the idea and does this represent an optimal situation for the parasite?

Accordingly, they are parasitized by reproductive and dispersing mites, respectively.

> Is redundant with previous sentence:

in an ectoparasite that exploits the adult stage of its host for dispersal and the developmental stages for reproduction.

Such host alternation between juvenile and adult stages of the same species is rare in ectoparasites and may require the advanced sociality with elaborate colonies and generation overlap that characterizes honey bees.

> Does it require sociality (and *V. destructor* could not exploit, say, solitary bees) or could it have been acquired thanks to sociality to increase fitness? Here a comparison with *Tropilaelaps* would be useful.

Methods:

The protonymphs, deutonymphs, and adult mites were collected 1-2 days prior to worker emergence, and stored at -80°C for further examination.

> A better description of the timing regarding host development is required to assess the internal morphology during the experimental period. The possibility to match this information with FigS3 and with the time at which mites stop ovipositing (ie reproducing) is required.

> I encourage the authors to discuss their results in the perspective of the recently published study by Piou et al.:

Piou, V.; Vilarem, C.; Blanchard, S.; Strub, J.-M.; Bertile, F.; Bocquet, M.; Arafah, K.; Bulet, P.; Vétillard, A. Honey Bee Larval Hemolymph as a Source of Key Nutrients and Proteins Offers a Promising Medium for *Varroa destructor* Artificial Rearing. *Int. J. Mol. Sci.* 2023, 24, 12443. <https://doi.org/10.3390/ijms241512443>

Dear Editor,

We would like to thank you and the reviewers for sending us the second round of reviews and giving us further opportunity to revise and improve the manuscript. We have carefully considered all comments and explain how we have taken them into account below.

Sincerely, Olav Rueppell (for all co-authors)

Reviewer #2 (Remarks to the Author):

Thank for your revised version of the manuscript. I found all comments to have been addressed, even though some responses were not integrated in the new version of the text, and are only present in the "response to reviewers" document.

Reply: Thank you for the positive assessment. We did only change the text in response to concerns that were general and kept some explanations that we considered direct responses to specific reviewer queries (and thus not of general interest) to the "response to reviewers" document.

Reviewer #3 (Remarks to the Author):

The authors have improved the text, which now presents the rationale much more clearly. Hypotheses are stated and the reader now understands what was done for which reason. Well done! The authors also added new data on *Tropilaelaps* proteomics and host internal morphology to support their arguments, this is remarkable.

Most my comments were addressed in a satisfactory manner. I nevertheless still have an issue with the main proteomics result as some choices made for comparing ingested proteins and reference hemolymph proteome do not appear justified to me. I also have some minor comments, which I hope can further improve the manuscript.

Reply: We thank the reviewer for appreciating our additional results and other efforts to improve the manuscript.

In absence of line numbering, I copy the manuscript text below and append my comments with a '>' sign under the respective text.

Reply: We apologize for the omission of line numbers and have now included them.

Major comments:

Overlap with a representative honey bee hemolymph proteome (50), was significantly higher ($\chi^2 = 10.3$, $p = 0.001$) for the proteins that were specific for foundresses (78/105 = 74.2%) than for proteins that were specific for dispersing mites (5/15 = 33.3%).

> This is a central argument, the whole proteomics approach and conclusion is based on this and thus needs to be given more prominence and clarity. I suggest making a figure to illustrate this result, which could complete Fig. 5a or replace Fig. 4. The latter is not very informative (because the more telling ratios are given in the text) and could be in the SM. The reader would also like to see a discussion on the relevance of this result for the research aim. Also a discussion about why it is not 100% and if a non 100% result is weakening your interpretation would be needed.

Reply: We agree that this is one important line of evidence. Thus, we have modified Figure 5a to illustrate the different portions of the Venn diagram with overlap to honey bee hemolymph proteome. The figure legend has been modified accordingly. We have also added to the discussion as suggested to highlight the importance of this result and explain why overlap is not 100% (there are technical and biological reasons that are now mentioned in the manuscript).

> Because this element was diluted among other results without being discussed, I previously failed to recognize its importance regarding the hypothesis tested. I now wonder why were only the proteins specific to either the dispersing mite or the foundresses considered for the match with hemolymph proteins and not the 167 proteins common to them, knowing that these are host proteins and not varroa proteins? Also host protein common to the two mite types should be considered to determine the food source, unless I missed something, which should be mentioned in the manuscript for other readers not to wonder as I did. Including these 167 proteins may shift the % match with hemolymph proteins between the mite types. In the extreme case of the 167 common proteins all belonging to the hemolymph, the % match become 90 for foundresses and 94 for dispersing mites!

Reply: We have now added the % match to hemolymph proteins in the “shared” proteins in the text and included this information also graphically in the figure. The suggested analysis of all identified proteins would still result in a significant difference (albeit with a more modest p-value of 0.0422). We did not include the shared proteins because we consider their inclusion methodologically problematic (because each of these proteins would be used twice in the statistical test) and we think that the contrast in hemolymph between the unique proteins the most meaningful test of our hypothesis.

> If I understand correctly, all the protein identified (272 for reproductive and 182 for dispersing individuals) are honeybee proteins (i.e. whole body proteome: hemolymph plus other tissues) and in the next step, only the hemolymph proteins are looked at. Why not restricting the comparison to the relevant hemolymph proteins directly? This would be less confusing. Would any important information be lost omitting whole body host proteome? From your argumentation in the text, I do not recognize any.

Reply: The proteomic analysis identified honey bee and mite proteins simultaneously: Two paragraphs below, we write “In addition to the honey bee proteins, our proteome comparison between foundresses and dispersing *V. destructor* identified 277 differentially expressed mite-endogenous proteins.” And in that paragraph, we derive some conclusions from the mite proteins that we consider important for the study.

It is currently impossible to restrict the analysis of honey bee proteins to hemolymph proteins a priori because the protein identification must be based on searching the raw mass spectrometry data, which includes all bee proteins. Secondly, the proportion of bee proteins that matches to hemolymph proteins identified in a different study is important (as pointed out by the reviewer above) and we cannot determine that proportion without taking all protein identifications into account.

Among the 78 proteins that were uniquely found in *Tropilaelaps*, 55 proteins (70.5%) overlapped with a representative honey bee hemolymph proteome (50).

> Why are only these *Tropilaelaps* specific proteins considered for a match with host hemolymph proteins? Those host protein also found in *V. destructor* if it feeds from the same

source also allow an interpretation regarding protein origin for *Tropilaelaps*. Considering the proteins also found in *V. destructor* might change the % match.

Reply: We agree with the reviewer that the proteins that were also found *V. destructor* are important and that is why we analyzed the overlap with the different *Varroa* protein groups in the first half of this paragraph (Fisher's exact $p < 0.0001$). But the reviewer is also correct that the overlap of the shared proteins with the hemolymph proteome is an important addition, which we have now included in the text.

> And same comment as above: why not focus straight away on hemolymph proteins?

Reply: See technical explanation above. Even if it was methodologically possible, we think that the overlap of *Tropilaelaps* to the different sets of *Varroa* proteins is better analyzed by including all bee-derived proteins to avoid biasing the results.

However, the quantitative difference (in hexamerins) to dispersing *V. destructor* demonstrates the utility of our proteome approach in general because the mite proteomes reflect the much higher expression of hexamerins in honey bee brood than in adults (60, 61).

> I see no obvious quantitative difference, if any at all, between dispersing and foundresses in Fig 5c. Are there statistics to support this claim?

Reply: There is an obvious difference because all of the normalized protein quantities for dispersing *Varroa* were negative while the foundress values were all positive. To better illustrate this difference graphically we have now created bar graphs that compare each protein for these groups side-by-side and depicted the Log₂ LFQ intensity values (without normalization) that were used in the statistical comparisons. The results of these comparisons are also now depicted in Fig. 5c and available in the supplementary tables.

Fig 5(C) Major royal jelly proteins and hexamerins from honey bees were consistently more abundant in foundresses than in dispersing mites.

> I see no consistent pattern here, see previous comment. Some MRJPs appear equally abundant in both groups. Are there statistics to support this claim?

Reply: See explanation above.

minor comments:

Title: It is a pity to restrict the work to *V. destructor* now that you completed the study with the previously missing *Tropilaelaps* proteomics. what about being more inclusive? Just a suggestion:

Life-history stage determines the diet of ectoparasitic mites on their honey bee hosts

Reply: We agree and have changed the title accordingly.

Abstract:

Biostaining and proteomic analyses were corroborated by corresponding experiments on *Tropilaelaps mercedesae*, a mite that only feeds on brood and has a strongly reduced dispersal stage.

> The ultimate aim is not to corroborate analyses, the aim of this comparison should be

rephrased as stated in the introduction.

Reply: We have changed the wording to give more weight to the *Tropilaelaps* results, but have refrained from adding too many words into the abstract.

Introduction

> What are direct experiments? This should be rephrased more accurately.

Reply: We have replaced “direct” with “decisive” to more accurately reflect our meaning.

Results and discussion

> Aims and hypothesis stating as well the mention of which methods used to reach the aims classically belongs to the introduction. The reader would also appreciate a mention of the secondary aims, which justify some results section on proteomics (physiological state of dispersing and foundress *V. destructor* with four figures) and metabolomics (speculative view into the developmental physiology), which are not directly linked to the main aim.

The results on feeding sites, however, do not fit in this section. Since the feeding site on pupae is known at least since Donze and Guerin 1994, your results confirming or precisizing this are not required to present the rationale of the next steps of your study.

Reply: We would like to remind the reviewer that this section was moved previously from the introduction, based on the first round of reviews. We don't think it should now be moved back. Furthermore, we realize that the feeding site characterization is confirming previous work but it is still original research that complements the other parts of our study, which is why we need to include it.

the majority of the life cycle of *V. destructor* and *Tropilaelaps* spp. consists of the reproductive stage during which these bee mites feed on pupating honey bees

> This statement is not totally correct because of long winters spent in the dispersal phase for *V. destructor*.

Reply: We replaced the wording to account for this technicality.

Our test used the same methodology of labeling host tissues with two fluorescent biostains, the hydrophilic Uranine and lipophilic Nile red, to distinguish the primarily hydrophilic hemolymph and lipophilic fat body in the mites after feeding (33).

> Same methodology as what?

Reply: We clarified that we refer to the methodology of the Ramsey et al. study that is cited at the end of the sentence (33).

Similar proteomic profiles of ingested food and specific host tissues provide molecular evidence of feeding preferences. We performed this analysis comparatively on reproductive and dispersing *V. destructor* to guard against potential biases that might favor the identification of proteins that are either derived from the honey bee fat body or hemolymph.

> the explanation for this precaution is better phrased in your rebuttal letter:

the potential bias we are guarding against would be a preferential representation of hemolymph- or fat body-derived proteins (e.g., one food type could be digested more easily than the other and therefore be less identifiable).

Reply: Thank you, we have added this explanation.

> it is however not clearer to me how the comparison between foundresses and dispersing individuals guards against this bias. Digestion ability could vary between life stages.

Reply: We agree in theory that we cannot exclude an interaction between mite life stage and food source digestibility (i.e., potential differences among host proteins could vary between dispersing and reproducing *Varroa*. We are just guarding against the potential bias that would arise from the more parsimonious scenario that different host proteins could have different digestibility regardless of *Varroa* life stage. All of this is hypothetical, because such biases are unknown, but we find it is better to guard against a potential bias that we can control. The interaction scenario invoked by the reviewer would indeed be very difficult to control.

> Fig 5(B) legend, 'non-significant' rather than 'insignificant'.

Reply: Changed.

(D) Differentially expressed proteins of *V. destructor*'s own proteome (non-honey bee derived) clustered perfectly between foundresses and dispersing mites.

> What is a perfect clustering? This should be rephrased.

Reply: We agree and have rephrased the statement.

> Fig. S3: how come the fourth column shows no green staining? These individuals have no hemolymph? The explanation becomes clear later in the methods:

The abdominal sternite was cut out and briefly washed with PBS. Then, the inner surface of the abdominal sternite was photographed

> In this case, no need to show or mention the Uranile column, what it is supposed to stain is gone!

Reply: We intend to show that the fat body is not quantitatively stained with Uranine and therefore prefer to keep the figure as is (also to be consistent with other figures).

The results indicated a high stability and reproducibility of our results because quality controls and other sample groups were tightly clustered together and separated from each other (Fig. 6A).

> 'tightly clustered together and separated from each other' sounds contradictory and would benefit from reformulation.

Reply: We agree and have rephrased that statement.

> Fig 6(C) legend, 'non-significant' rather than 'insignificant'.

Reply: Changed.

although protonymphs, deutonymphs, and foundresses of *T. mercedesae* appeared not as different from each other as experimental groups of *V. destructor*.

> This observation deserves a discussion, even if speculative: why is it so?

Reply: We discuss possible reasons after presenting (and statistically evaluating) the quantitative data in the next few sentences. The discussion starts with the phrase "This considerable difference may be attributed to...." We have extended that discussion to address the next comment.

However, these ratios in *T. mercedesae* varied less and exhibited a pattern that was inverse of what was observed in *V. destructor* with the highest ratio in protonymphs and the lowest in foundresses.

> Same comment as above. Below you only discuss the large differences in absolute values between the species.

Reply: See above.

Conclusion:

Our findings are incompatible with the claim that *V. destructor* rely on honey bee fat body (33)

> This statement should be nuanced by adding 'primarily' and/or by specifying 'during all life stages'.

Reply: We agree and have added "primarily" as suggested.

The dispersal, reproductive, and developmental stages have distinct nutritional needs

> This may be true, but your work indicates that it is not the need that determines the source of the food, but its accessibility. See your explanation on why the less nutritive hemolymph suffices to sustain reproduction. See also next comment. Is there a clinch or compromise here or not? This is unclear to me.

Reply: We have modified this sentence and added another sentence to clarify our meaning here, which is indeed arguing (carefully) for food accessibility as the main driver of *Varroa* feeding behavior.

These host stages match *V. destructor* requirements for mobility with differences in food availability because the brood is immobile and filled with proteinaceous hemolymph, while the adults are highly mobile and contain more localized fat body tissues.

> Please better explicit the connection between mobility and food type. Proteinaceous hemolymph sustain reproduction metabolism and fat body tissues energy metabolism? Is this the idea and does this represent an optimal situation for the parasite?

Reply: We did not mean to imply this and have reformulated the paragraph.

Accordingly, they are parasitized by reproductive and dispersing mites, respectively.

> Is redundant with previous sentence:

in an ectoparasite that exploits the adult stage of its host for dispersal and the developmental stages for reproduction.

Reply: Yes, we agree and have deleted this sentence.

Such host alternation between juvenile and adult stages of the same species is rare in ectoparasites and may require the advanced sociality with elaborate colonies and generation overlap that characterizes honey bees.

> Does it require sociality (and *V. destructor* could not exploit, say, solitary bees) or could it have been acquired thanks to sociality to increase fitness? Here a comparison with *Tropilaelaps* would be useful.

Reply: We thank the reviewer for suggesting to consider *Tropilaelaps* and have modified the paragraph accordingly.

Methods:

The protonymphs, deutonymphs, and adult mites were collected 1-2 days prior to worker emergence, and stored at -80°C for further examination.

> A better description of the timing regarding host development is required to assess the internal morphology during the experimental period. The possibility to match this information with FigS3 and with the time at which mites stop ovipositing (ie reproducing) is required.

Reply: We have added this information to allow the direct comparison to Fig. S3.

> I encourage the authors to discuss their results in the perspective of the recently published study by Piou et al.:

Piou, V.; Vilarem, C.; Blanchard, S.; Strub, J.-M.; Bertile, F.; Bocquet, M.; Arafah, K.; Bulet, P.; Vétillard, A. Honey Bee Larval Hemolymph as a Source of Key Nutrients and Proteins Offers a Promising Medium for Varroa destructor Artificial Rearing. Int. J. Mol. Sci. 2023, 24, 12443. <https://doi.org/10.3390/ijms241512443>

Reply: Thank you, we have now included this recent article into our discussion.

REVIEWERS' COMMENTS

Reviewer #3 (Remarks to the Author):

The authors have addressed all my comments in a satisfactory manner and I have no further remark.